# High-throughput deconvolution of 3D organoid dynamics at cellular resolution for cancer pharmacology with Cellos

Patience Mukashyaka [1,2,3], Pooja Kumar[1,3], David J. Mellert [1], Shadae Nicholas[1], Javad Noorbakhsh[1], Mattia Brugiolo[1], Elise T. Courtois [1], Olga Anczukow[1,2], Edison T. Liu [1] ✉ & Jeffrey H. Chuang [1,2] ✉

Three-dimensional (3D) organoid cultures are flexible systems to interrogate cellular growth, morphology, multicellular spatial architecture, and cellular interactions in response to treatment. However, computational methods for analysis of 3D organoids with sufficiently high-throughput and cellular resolution are needed. Here we report Cellos, an accurate, high-throughput pipeline for 3D organoid segmentation using classical algorithms and nuclear segmentation using a trained Stardist-3D convolutional neural network. To evaluate Cellos, we analyze ~100,000 organoids with ~2.35 million cells from multiple treatment experiments. Cellos segments dye-stained or fluorescently-labeled nuclei and accurately distinguishes distinct labeled cell populations within organoids. Cellos can recapitulate traditional luminescence-based drug response of cells with complex drug sensitivities, while also quantifying changes in organoid and nuclear morphologies caused by treatment as well as cell-cell spatial relationships that reflect ecological affinity. Cellos provides powerful tools to perform high-throughput analysis for pharmacological testing and biological investigation of organoids based on 3D imaging.

The selection of an optimal disease model is critical to the effective development and evaluation of cancer therapeutics. 2D monolayer cell cultures have been used extensively but have notable limitations. For example, although they can model cell autonomous characteristics, they lack multicellular aspects of the in vivo environment[1] that may affect therapeutic performance, such as contributions of stromal components, spatial architecture, and cell polarity. Moreover, adaptation of cancer cells to adherent conditions on artificial substrates often perturb cellular characteristics. For such reasons, 3D cell culture models have emerged as a high-scale in vitro model platform for anticancer therapeutic discovery and development[2–6]. 3D profiling can enable broad analysis possibilities because, unlike 2D profiling, it captures true cell spatial relationships and morphology, as well as potential cell-cell interactions. However, to realize these possibilities for high-throughput

treatment testing, improved 3D organoid data analysis is a key need[7].

Organoid culture analysis is typically performed through image capture of organoids grown in multi-well plates, but available methods have limitations. Luminescence assays that are commonly used are cell-destructive methods that aggregate cell growth information over an entire well, whereas analysis of individual organoids and their constituent cells would be more informative. Current methods to identify organoids have focused on 2D segmentation, e.g. based on planar fluorescent quantification of live/dead cell stains and organoid areas[8–14]. Yet 2D analysis poorly approximates the complexity of 3D spatial relationships for cells and organoids[14], which are likely to vary by cell type and growth conditions.

Moreover, prior computational approaches to analyze organoids in 3D have focused on qualitative visualization[7] rather than cellular

[1]The Jackson Laboratory for Genomic Medicine, Farmington, CT, USA. [2]Department of Genetics and Genome Sciences, University of Connecticut Health Center, Farmington, CT, USA. [3]These authors contributed equally: Patience Mukashyaka, Pooja Kumar. ✉e-mail: ed.liu@jax.org; jeff.chuang@jax.org

quantification, and current methods for 3D cell segmentation have been limited to specialized contexts. For example, Boutin et al.[15] developed a 3D spheroid and nuclei segmentation approach for optically cleared images of one single spheroid in a well. More recently, Beghin et al. developed a segmentation technique for the Jewell system again with one organoid per well[16]. However, a more typical cancer treatment assay interrogates large numbers of organoids within each well of a plate. Related to this need, Zhang et al.[17] described a method optimized for segmenting organoids in optical coherence tomography images, though the method lacks cellular resolution. Thus, development of a flexible 3D approach able to identify and quantify individual cells, as well as their morphologies, in high-throughput could be of significant value to the field.

We demonstrate a computational method, Cellos (Cell and Organoid Segmentation), to address this challenge. Cellos performs high-throughput volumetric 3D segmentation and morphological quantification of organoids and their cells on images with numerous organoids. Cellos has two stages. At first, organoids are segmented, and their volume, solidity, and other morphological characteristics are computed. In the second, nuclei are segmented in each organoid using a 3D convolutional neural network trained on an extensively curated dataset. This enables analysis of characteristics including cell densities per well, clonal population frequencies, nuclear morphologies, and potential cell-cell interactions. We demonstrate the utility of Cellos by quantifying the complex three-dimensional responses of triple negative breast cancer (TNBC) organoids and their subclonal populations during platinum-based treatment in vitro.

## Results

### Description of the cancer cellular system

As a case study for 3D image analysis, we generated organoids from primary cell cultures derived from a TNBC Patient Derived Xenograft (PDX) model TM00099[18]. We used these cells because they have not been adapted to 2D cell culture conditions and retain clonal heterogeneity, and thus resemble the clinical in vivo situation. This TNBC model is *BRCA1* deficient and was initially sensitive to cisplatin. We have previously reported[18] that this PDX tumor consists of two major subclones with differential cisplatin sensitivity. We isolated and established single cell-derived clonal lines, a cisplatin-resistant clone A50 and relatively sensitive clone B. We performed $IC_{50}$ measurements for each of these clones individually grown as 3D organoids using a standard cell destructive luminescence assay, which assesses the number of viable cells from ATP content after cellular disruption (see methods for details). The A50 clone had an $IC_{50}$ of 10.51 μM (95% CI = 8.71 μM–12.44 μM) and the relatively sensitive B clone had an $IC_{50}$ of 2.94 μM (95% CI = 2.61 μM–3.35 μM) (Fig. 1a). However, at high concentrations of cisplatin ($>IC_{80}$), subclone B also exhibited a "dormancy" phenotype with elevated cell survival over A50. We took advantage of these clones to test the effectiveness of Cellos to quantify mixed populations with complex sensitivity profiles. We used nuclear localization signal (NLS)-conjugated EGFP or mCherry to stably label A50 and B clones and mixed them in two different ways: *homogeneously mixed organoids*, defined to be organoids made up of two genetically identical but differently labeled subclones (A50-EGFP with A50-mCherry, or B-EGFP with B-mCherry); and *heterogeneously mixed organoids*, defined to be organoids made up of two different subclones (A50-EGFP with B-mCherry, or A50-mCherry with B-EGFP). Organoids were imaged using the PerkinElmer Opera Phenix high-content screening system (Supplementary Fig. 1).

### Cellos pipeline overview

The pipeline consists of two parts: organoid segmentation and nuclei segmentation (Fig. 1b). *Organoid segmentation*: To segment the organoids, first, we convert the fluorescent image to grayscale and pre-process to remove debris and noise (see Methods for details). Next, we use the Triangle method for histogram thresholding[19] to create a binary image separating the organoids from the background. We then use scikit image[20] to uniquely label all organoids, remove small objects (Fig. 1c), and generate a table of measurements for the remaining organoids (3D bounding box, volume, mean intensity, solidity, etc.). All steps are performed on small fields defined by the imaging platform used and then stitched together before the labeling step. A csv file that contains measurements of all organoids is generated for each well in the plate, allowing parallel processing of individual wells from the same plate. *Nuclei segmentation*: For each well, stitched z-stack images and the measurement file for each individual organoid are used as inputs. Segmentation of nuclei within the organoids by classical methods is challenging due to the high density of cells in organoids. Therefore, we developed a convolutional neural network (CNN) for nuclei segmentation using the Stardist-3D[21] model with a ResNet backbone[22]. To train the model, we generated a training dataset of 3862 manually annotated nuclei in 3D from 36 TNBC organoids, with a range of 8-440 cell nuclei per organoid. The organoids consisted of EGFP, mCherry or Hoechst labelled cells, and 24 of the 36 organoids were imaged after exposure to a range of cisplatin concentrations. We then applied the trained model to experimental data (Fig. 1d). We also used scikit-image to generate measurements (centroid, volume, mean intensity, solidity, etc.) for every nucleus. Our trained CNN is publicly available on the Cellos GitHub.

### Cellos organoid segmentation is accurate

To evaluate the effectiveness of Cellos for segmenting organoids, we applied it to heterogeneously mixed organoids (A50-EGFP with B-mCherry) in several wells. These wells were treated with cisplatin concentrations ranging from 0-128 μM (see methods experiment-1 for details). Organoid segmentation appeared accurate in the absence (Fig. 2a, left panel) and presence (Fig. 2a, middle and right panels) of cisplatin. For more organoid segmented images, see Supplementary Figs. 2−5a−c. To quantify this, we manually counted true positives, false positives, and false negatives for 321 segmented organoids in wells treated with different concentrations of cisplatin (16-128 μM). We focused on high cisplatin doses because they are more difficult to accurately segment−their organoid shapes are more diffuse, and the images have higher cell debris and noise. We observed precision, recall, and F1 scores of 96.07, 83.80, and 89.52, respectively, where the F1 score is the harmonic mean of recall and precision. At high doses of drug, the cutoff on minimum organoid size caused the recall value for organoid segmentation to be lower than the precision. The cutoff, which was chosen to reduce the effect of debris, also reduced recall by excluding some single cells from being segmented as organoids, as shown in Supplementary Fig. 6.

### Cellos nuclei segmentation model is accurate

To evaluate Cellos for nuclear segmentation, we first used internal cross-validation. We performed six-fold cross-validation (30 and 6 organoids for training and validation respectively) and computed F1 score versus intersection over union (IoU) thresholds. IoU is the spatial overlap between the ground truth and the predicted nuclear region− the bigger the IoU, the greater the overlap. We observed high F1 scores extending to relatively large IoU thresholds, for example an F1 score of 0.853 ± 0.052 at IoU of 0.4 (Fig. 2b), indicating the quality of the match between predicted nuclei and ground truth. Visual inspection supported the consistency in ground truth and predicted labels, and the total numbers of nuclei identified were similar in the predictions and ground truth (Fig. 2c). For additional visuals of nuclei segmentation, see Supplementary Figs. 2−5d. These results show the accuracy of Cellos to segment nuclei, despite the challenges of high anisotropy in z resolution relative to x and y[21] and increased debris after exposure to high concentrations of drug.

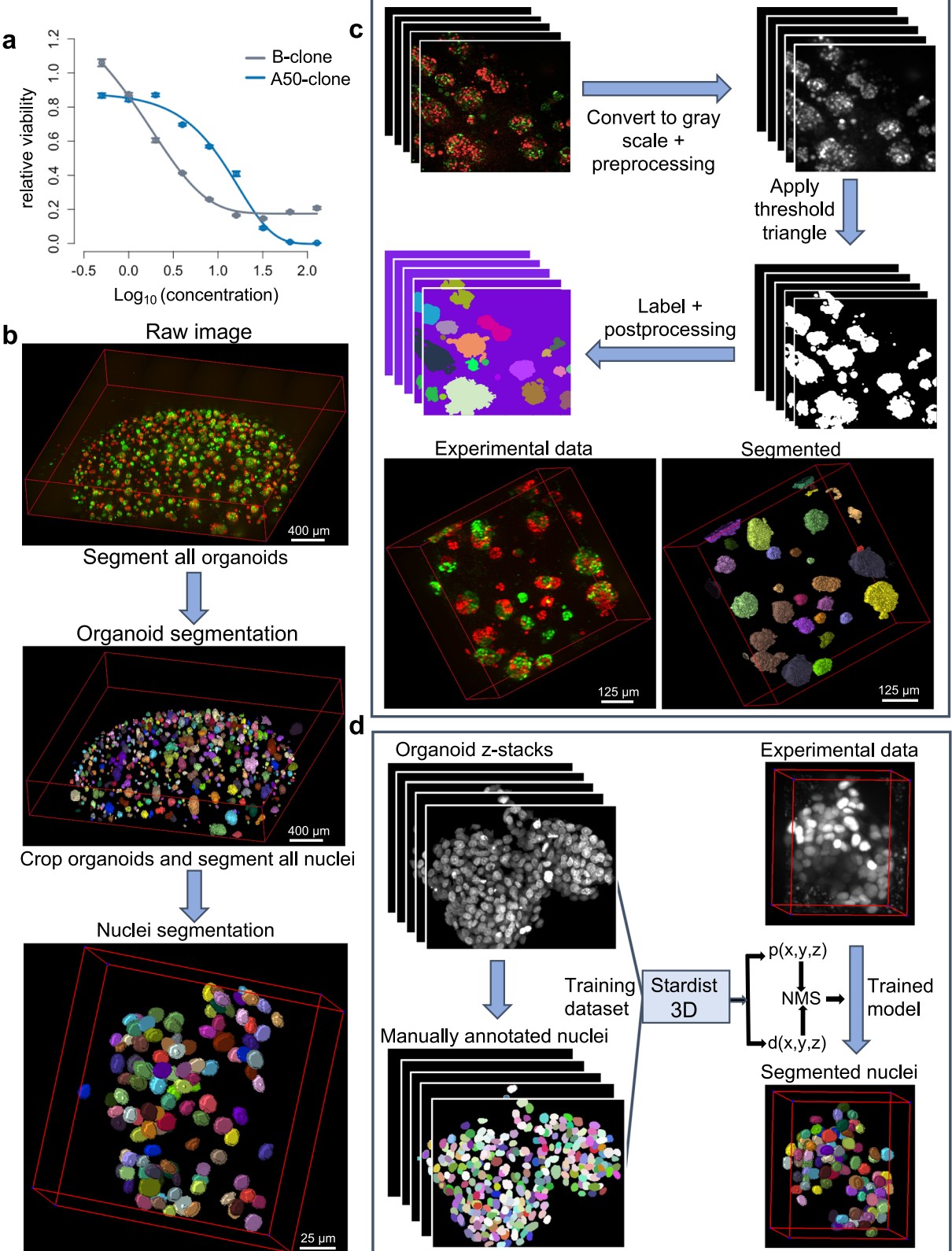

**Fig. 1 | Outline of *Cellos* pipeline and cellular system. a** Cisplatin $IC_{50}$ curves for two clones A50 (blue line) and B (gray line) from 3D homogeneously mixed organoids using a cell-destructive luminescence readout. Mean of three replicate wells is plotted and error bars represent the standard deviation. Source data is provided as a source data file. **b** Cellos: Two-stage pipeline for 3D organoids and nuclei segmentation on 3D images. Scale bar represents 400 μm for top and middle panel and 25 μm for the bottom panel. **c** Top panel represents steps for 3D organoid segmentation. The inputs are 3D z-stack images, and the outputs are the segmented and labeled organoids. Bottom panel shows an example of 3D organoids before and after segmentation. Scale bars shown represent 125 μm. **d** Steps for nuclei segmentation. A Stardist-3D with Resnet backbone model[22] is trained using the training dataset. The trained model is then applied to experimental data with individual segmented organoids as input, and segmented and labelled nuclei as outputs. The 3D figures were generated using the napari[46] python library, which is integrated into the Cellos pipeline.

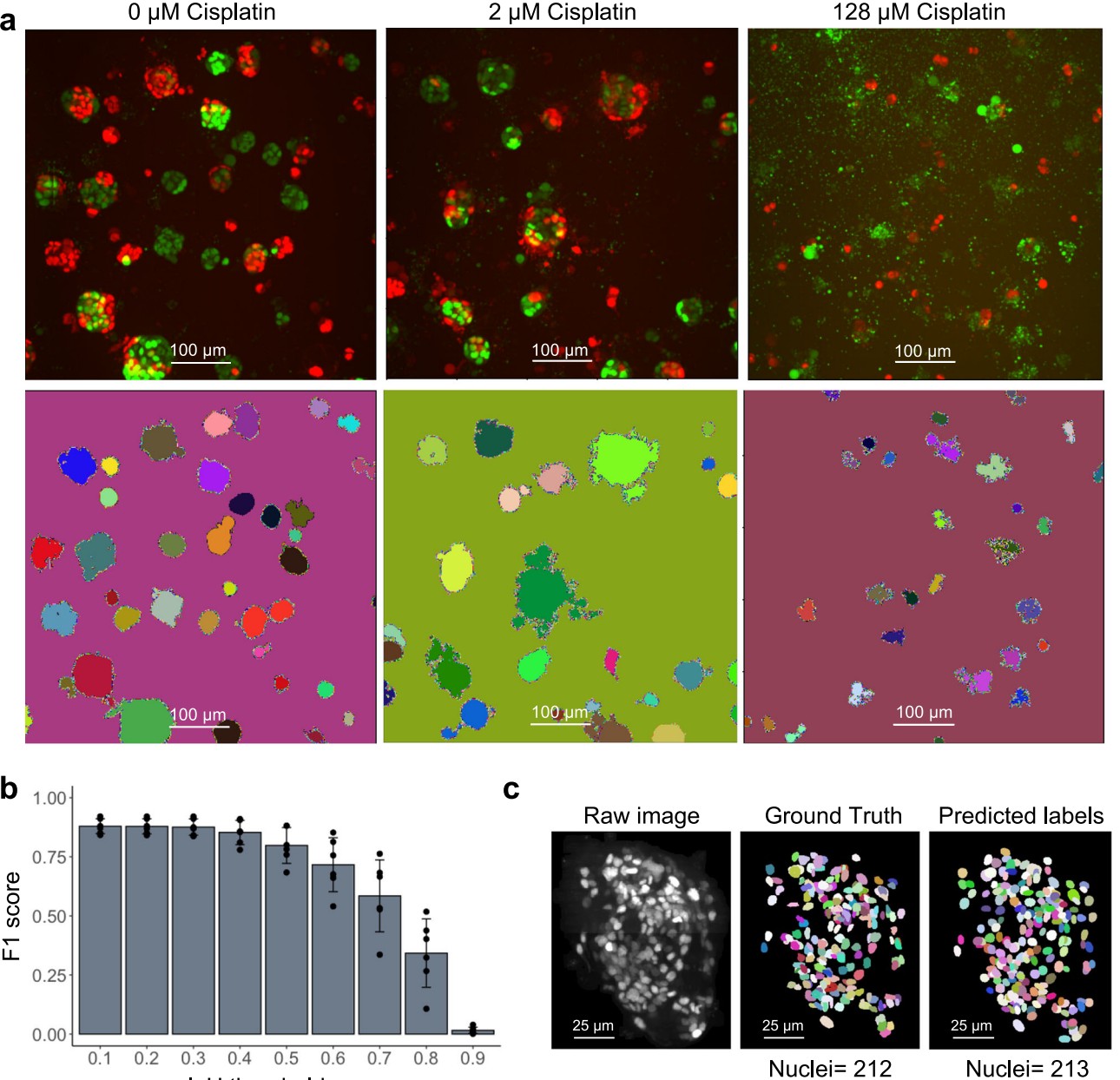

**Fig. 2 | Evaluation of organoid and nuclei segmentation. a** For each pair of images, the top panel shows fluorescence *z*-axis maximum projections with A50 cells labeled with EGFP and B cells labeled with mCherry. The bottom panel shows organoids segmented by Cellos. Segmented individual organoids are in distinct colors. Organoids from untreated, 2 μM, and 128 μM cisplatin wells are shown and scale bar represents 100 μm. Representative segmentation for one field per condition is shown. Segmentation was performed on 25 fields per well and three replicate wells for each condition. **b** F1 score for nucleus identification vs. IoU threshold. Mean and standard deviation across six cross-validations are shown. A total of 3,862 nuclei were used for this analysis. Source data is provided as a source data file. **c** Example of EGFP-labeled organoid image (left panel) with manually annotated ground truth nuclei annotations (middle panel) and Cellos predicted labels (right panel), respectively. Images are *z*-axis maximum projections and scale bar represents 25 μm.

## Cellos organoids and nuclei segmentation is robust on independent datasets

To test the broader applicability of Cellos, we determined if Cellos was effective on organoids from cell lines with different organoid morphologies. For this purpose, we analyzed a total of 426,810 cells from 11,416 organoids generated from TNBC cell lines HCC1806 and MDA-MB231 that have mass-like and stellate morphologies, respectively, as well as breast cell line MCF10A that has round and cyst-like organoids when differentiated (Fig. 3a–c). In this challenge, Cellos successfully segmented the organoids with distinct morphologies in a quantitative manner. In all cases, we observed a decrease in volume of segmented organoids with decreasing seeding densities of cells

(Supplementary Fig. 7a). Specifically, for HCC1806 the average organoid volumes ($\times 10^5$ μm³) were $2.68 \pm 0.01$, $2.65 \pm 0.07$, and $2.17 \pm 0.07$ for high, medium, and low cell seeding densities respectively. For MCF10A the average organoid volumes ($\times 10^5$ μm³) were $1.50 \pm 0.20$, $1.39 \pm 0.21$, and $0.96 \pm 0.24$ for high, medium, and low cell seeding densities respectively. For MDA-MB231 the average organoid volumes ($\times 10^5$ μm³) were $2.99 \pm 0.52$, $2.59 \pm 0.48$, and $1.89 \pm 0.28$ for high, medium, and low cell seeding densities. Additionally, as expected, the number of cells per organoid segmented using Cellos decreases as the seeding density decreases, with $p < 0.022$ in all comparisons between high and medium seeding densities ($p < 2.551e{-}08$ in all comparisons, Supplementary Fig. 7b). Thus, we observed that Cellos effectively

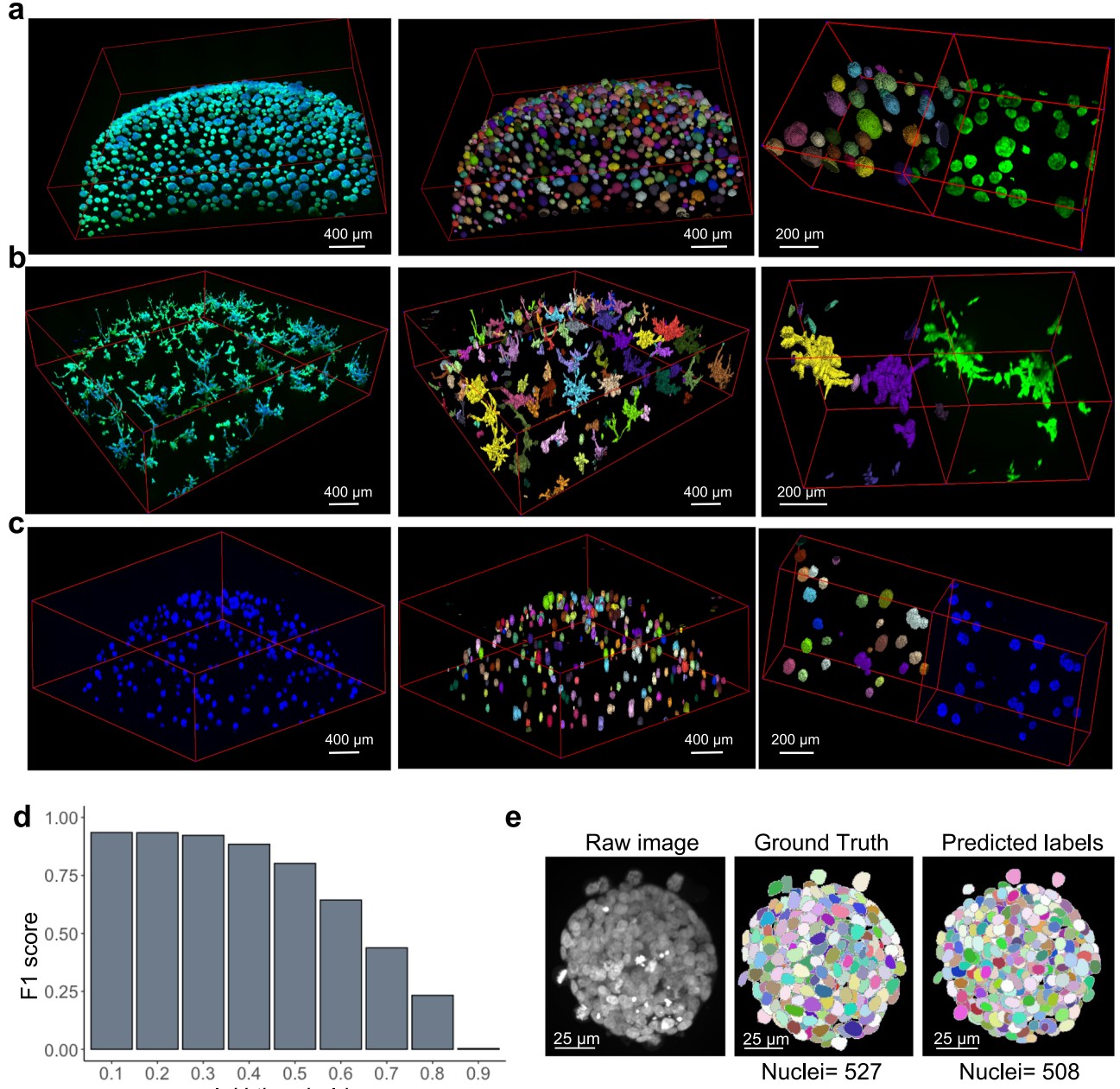

**Fig. 3 | Application of Cellos on diverse organoid datasets.** 3D images of **a** HCC1806, **b** MDA-MB231 and **c** MCF10A organoids. Raw images are shown in the left panel, organoid segmentations via Cellos are shown in the middle panel, and a smaller field for better visualization of organoid segmentation is shown in the right panel. Individual segmented organoids are displayed in different randomly selected colors. Segmentation was performed on at least three replicate wells for each cell line and representative segmentation for one well per condition is shown. Scale bars represent 400 or 200 μm as indicated. **d** F1 score of Cellos nuclei segmentation in spheroids of breast carcinoma from Boutin et al.[15] vs. IoU threshold. A total of 1585 nuclei was used for the analysis. Source data is provided as a source data file. **e.** Example DAPI stained image (left panel) from Boutin et al.[15] with ground truth (middle panel) and Cellos predictions (right panel), respectively and scale bar representing 25 μm.

segments nuclei from organoids with distinct morphologies. Moreover, despite the variations in organoid morphologies and fluorescent dyes used for segmentation (see methods), no changes in Cellos pipeline parameters were required.

We next tested whether Cellos could segment nuclei from organoids in external datasets generated independently from our group. We used data from Boutin et al.[15], who manually annotated cells in three optically cleared spheroids grown and assayed in a manner distinct from our platform, which does not use optical clearing. The spheroids were from a breast carcinoma cell line T47D grown in 384-well round-bottom Ultra-Low Attachment plates for 3 days. They were stained with DAPI and imaged using a 20x water objective magnification with 5 μm z-step size. Despite these system and fluorescence marker differences, Cellos accurately segmented the nuclei (1585 nuclei) in their spheroids with $F1 = 0.885$ at IoU of 0.4 (Fig. 3d), with an example spheroid shown in (Fig. 3e). We compared Cellos to the 3D classical nuclei segmentation protocol that Boutin et al. developed for their spheroids. Cellos showed superior specificity of 93.38% and $F1 = 0.94$ compared to the Boutin et al. method with 75.76% for specificity and $F1 = 0.76$.

Additionally, we applied Cellos to a leukemia cell line HL60 synthetic dataset[23], consisting of four synthetic image subsets

each containing 30 images at different nuclei densities and signal to noise ratios. Cellos was able to segment the nuclei with precision, accuracy, and recall of >0.95 in all subsets. Finally, we were also able to segment 52 out of 56 annotated nuclei from an image of a mouse embryo[24]. Thus, Cellos was able to process external image datasets with quantitative precision.

## Cellos precisely detects distinct proportions of labelled cells in mixed organoids

We wished to evaluate the efficacy of Cellos to quantify multiple cell types simultaneously in an organoid to study cancer population dynamics. To do so, we first analyzed the ability to distinguish mixed organoids expressing different fluorescent tags using the PDX TM00099 A50 clone. Organoids were made up of mixtures of A50-EGFP and A50-mCherry, at four different seeding ratios of EGFP-20%, EGFP-40%, EGFP-60%, and EGFP-80%. These organoids were seeded on day "-3" and imaged at day 0 (days are counted relative to when organoids were stably formed, Supplementary Fig. 8a), allowed to grow, and imaged again 4 days later (day4) (see methods experiment-2 for details). We performed similar experiments for homogeneously mixed fluorescent organoids of TM00099 clone B-EGFP with B-mCherry.

To evaluate this quantitatively, we used Cellos to calculate the total number of EGFP and mCherry cells in each well and their corresponding ratios. We observed that the EGFP fluorescence increased with the EGFP seeding ratio in organoids made of A50-EGFP with A50-mCherry (Fig. 4a). Indeed, across all wells imaged at day0, we detected ratios close to the expected EGFP:mCherry ratios, with average absolute difference of 2.986% (Fig. 4b). Ratios were also stable from day0 to day4, with mean difference of 2.852%. We calculated the standard deviation within each set of triplicate wells, and the average deviation across seeding conditions was 0.808% (Fig. 4b). We repeated these experiments for organoids generated from mixtures of B-EGFP and B-mCherry and found similar results (Fig. 4b). We did not detect bias towards either fluorescent label or clone type (Supplementary Table. 1).

Next, we evaluated how these ratios varied within individual organoids across the different seeding conditions. Using Cellos, we quantified the number of EGFP and mCherry cells in each organoid in each well (total 561,722 nuclei and 23,258 organoids) and plotted these values across organoids (Fig. 4c) for each seeding condition. As expected, the slopes of the regression line between number of EGFP versus mCherry cells changed monotonically with seeding ratio. Specifically, we observed slopes of 2.270, 1.347, 0.814, 0.427 for the wells with seeding conditions of EGFP-20%, 40%, 60%, and 80%, respectively for A50 (Fig. 4c). Observations were similar for B (Supplementary Fig. 8b). This indicates that Cellos is able to discern cellular ratios across different organoid sizes.

## Cellos robustly quantifies treatment response in co-cultured clones

An important use of Cellos is to quantify treatment response profiles of organoids made up of multiple clones, a task which may be valuable for parallelizing response evaluations and detecting clonal interactions. The A50 and B clones from TM00099 tumors have distinctive drug responses to cisplatin (Fig. 1a) making them well-suited to evaluating Cellos for this task because of the differential sensitivity to chemotherapy. We generated heterogeneously mixed organoids consisting of A50-EGFP and B-mCherry cells and treated them with a range of (0-128 μM) cisplatin concentrations (see methods experiment-1) in triplicates. We first used Cellos to estimate cell density in each well. As expected, we saw a decrease in cell density as cisplatin concentration increased (Fig. 5a), confirming effectiveness of treatment. We then used clone-specific cell densities to determine $IC_{50}$ for each clone. A50 was 2.7× more resistant to cisplatin than B, i.e. A50 $IC_{50}$ = 2.87 μM (95%

confidence interval (CI) = 1.95 μM–4.54 μM) and B $IC_{50}$ = 1.06 μM (95% CI = 0.87 μM–1.35 μM) (Fig. 5b). At higher doses (~$IC_{80}$), and as expected, we observed a higher viability of B than A50. These observations were consistent with the standard luminescence assays (Fig. 1a), indicating that Cellos can recapitulate those findings by counting individual fluorescently labelled nuclei of specific cell types, but in a nondestructive and higher (cell) resolution manner.

To further visualize these results, we analyzed the ratio of A50 and B cells in the cisplatin-treated wells containing heterogeneously mixed organoids. For each drug condition, the ratio of A50-EGFP to B-mCherry cells per well was computed (Supplementary Fig. 9a) and normalized relative to the ratio in untreated wells. We observed increasing A50-EGFP and decreasing B-mCherry normalized cell proportions with increasing cisplatin concentration, up to 16 μM (Fig. 5c). This trend reversed at higher concentrations (64 and 128 μM), consistent with the luminescence and Cellos-based cell density estimates. This result highlights the dormant phenotype of B having better survival than A50, but only at high doses of cisplatin.

To determine whether fluorescence labeling might have impacted cell viability or generated any imaging bias, we flipped the fluorescent labels and repeated the experiment, i.e. generating and treating heterogenous organoids with A50-mCherry and B-EGFP cells (see methods, experiment-3). The flipped-label results were highly correlated with the originals (R = 0.982, Fig. 5d, see also Supplementary Fig. 9b), confirming that the observed treatment responses were determined by intrinsic clonal differences. We also analyzed homogeneously mixed organoids consisting of A50-EGFP and A50-mCherry cells treated with cisplatin (see methods, experiment-4). As expected, the normalized proportions of EGFP and mCherry cells were stable across all cisplatin conditions (Fig. 5e). $IC_{50}$ and cell density curves showed variability across replicates related to the variable spatial distribution of organoids relative to the imaged regions in each well. However, the ratio of EGFP to mCherry was highly consistent across replicates (Fig. 5c), demonstrating the improved robustness of measures based on internal calibration, compared to measures based on absolute cell counts.

At the individual organoid level, we observed similar clonal compositional shifts and were able to quantify the clonal variation on a per organoid basis (Fig. 5f, g). The median percentage of A50 EGFP cells per organoid in control untreated wells was 57%. This increased to 72% at 2 μM cisplatin (p = 2.774e–13) and decreased to 23% at 64 μM (p = 3.142e–56, Fig. 5f). Though consistent with the bulk behavior of A50 vs B clones, for organoids with few cells we observed wider variation in the A50-EGFP percentage, as expected from their susceptibility to stochastic fluctuations (Fig. 5g). When we computed the clonal ratios using mean fluorescence intensities within each organoid, we observed findings similar to those from the cell counts (Supplementary Fig. 9c).

To determine if organoid and nuclear segmentation post drug treatment would be efficient in a different cell line model and using different therapeutic agents, we used Cellos to analyze organoids generated from TNBC cell line HCC1806 treated with cisplatin or two additional chemotherapeutic agents commonly used for TNBC treatment, namely Docetaxel and Doxorubicin (Supplementary Fig. 10a). Cellos was used to segment HCC1806 organoids and yielded a monotonic decrease in organoid volume as the drug concentration increased for all three drugs (Supplementary Fig. 10b). Additionally, the number of viable nuclei counted using Cellos was highly correlated with luminescence assay signals acquired post imaging (correlation = 0.961, 95% confidence interval = 0.916–0.982) (Supplementary Fig. 10c). Cellos was effective in organoid and nuclear segmentation despite the organoids being treated with relatively high drug doses, which yields abundant cell debris that in principle could have interfered with segmentation (Supplementary Fig. 10a). Note that segmented nuclei that were

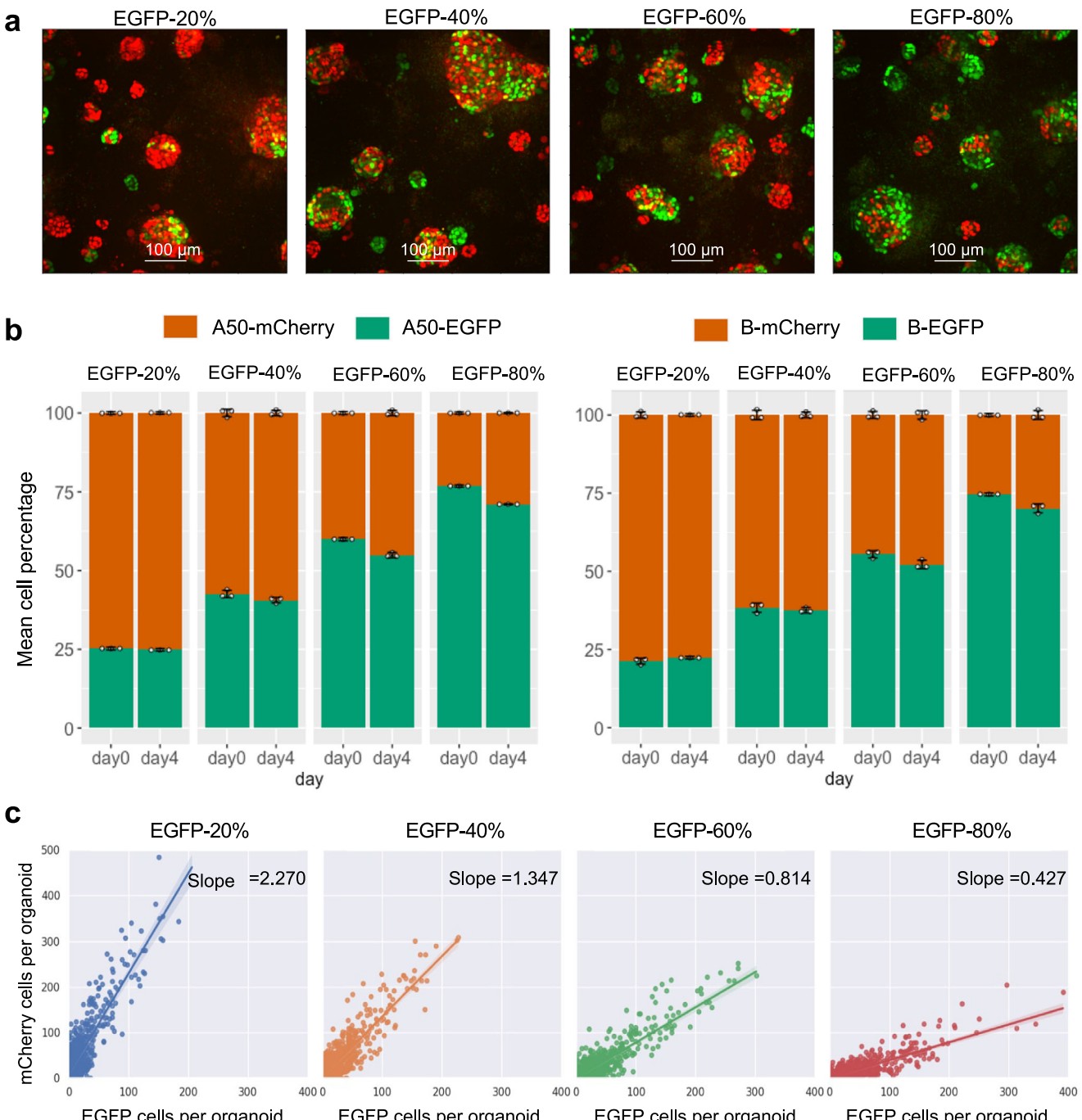

**Fig. 4 | Cellos distinguishes proportions of co-cultured cells. a** Representative z-axis maximum projection images of homogeneously mixed organoids generated with seeding percentages of 20%, 40%, 60% and 80% A50-EGFP, respectively, with the remaining cells being A50-mCherry. Images shown are from day4. Twenty-five fields per well and three replicate wells for each condition were analyzed and representative images for one field per condition is shown. **b** Stacked bar plot showing the percentage of A50-EGFP+A50mCherry cells detected by Cellos at day0 and day4 for each of the seeding conditions. Analogous experiments for B-EGFP + B-mCherry mixed organoids are shown in the right panel. Mean of three replicate wells are plotted for each condition, white points show the data for individual replicates wells and error bars indicate the standard deviation. A total of 1,123,444 cells were analyzed. **c** Number of cells labelled with EGFP vs. mCherry detected in each homogeneously mixed A50 organoid. Each dot depicts an organoid Seeding conditions of EGFP-20% (blue), EGFP-40% (orange), EGFP-60% (green) and EGFP-80% (red) are shown from left to right. In total, 3245 organoids with a total of 185,702 cells were analyzed. The slope of the fitted linear regression is noted with the shaded bands indicating the 95% confidence interval. Source data for **b** and **c** are provided as source data files.

also positive for dead cell stain DRAQ7 were identified as dead cells and removed from the analysis. These results show the ability of Cellos to segment organoids and nuclei generated from an established cell line treated with three drugs with distinct mechanisms of action.

## Cellos can reveal organoid and nuclear morphology changes due to therapy

Organoid morphology may change in response to therapy and can be used to evaluate effects of drugs on cancer organoids in qualitative image-based assays[25–27]. We therefore investigated whether Cellos

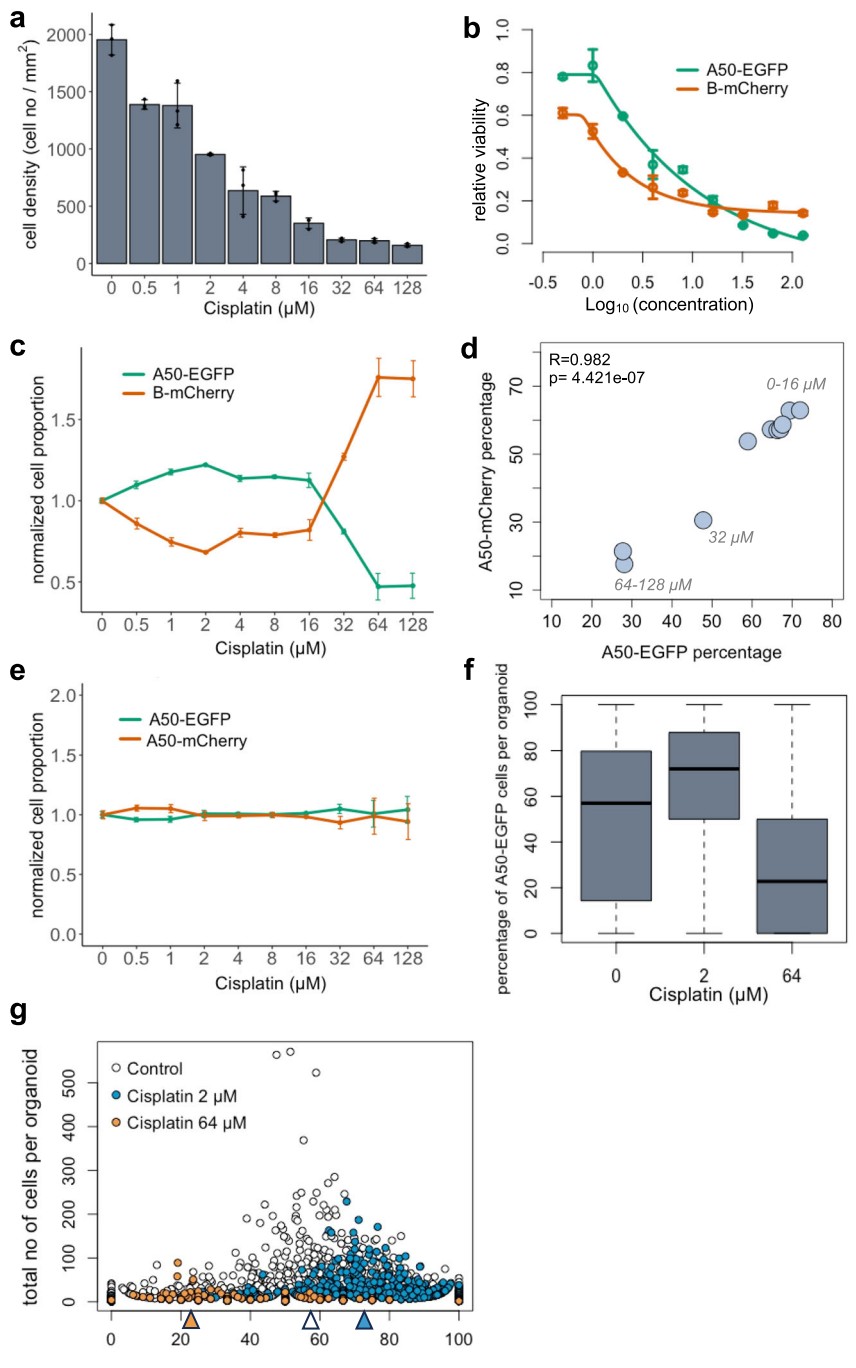

**Fig. 5 | Cellos quantification of treatment response for organoids consisting of co-cultured clones. a** Total cell density (for both A50 and B) after exposure to cisplatin (0-128 μM) for 4 days. Mean and standard deviation across three replicate wells per condition are shown and data for individual replicates are shown as black dots. **b** IC₅₀ curves for A50-EGFP and B-mCherry clones when co-cultured in heterogeneously mixed organoids. Mean values with standard deviation across triplicates indicated for each condition. **c** Control normalized cell proportions vs. cisplatin treatment concentration range, for A50-EGFP and B-mCherry clones cultured as mixed organoids. Line plots shows mean and standard deviation of three replicates for all conditions. A total of 137,765 cells were examined for (**a**–**c**). **d** Correlation of A50 clonal percentages post cisplatin treatment, when labeled with either nuclear EGFP or mCherry and mixed to form organoids with B clones that had been labelled with the other fluorescent channel. 238,077 cells were utilized for this analysis. Pearson correlation coefficient and two-sided *t*-test *p* value are noted. **e** Clonal cell proportions for A50-EGFP and A50-mCherry clones co-cultured as mixed organoids, normalized by proportion in the untreated control, as a function of cisplatin treatment concentration. Data was collected from three replicates for all but one condition that had two replicates available. A total of 123,069 cells were assessed. Mean values with standard deviation are plotted. **f** Distribution of percentage of A50-EGFP cells per organoid when mixed with B-mCherry cells after 0, 2 or 64 μM cisplatin exposure. Horizontal line in the boxplot indicates the median, the box denotes the interquartile range (IQR) and the whiskers beyond the box extend to a maximum of 1.5 times the IQR. **g** Percentage of A50-EGFP cells per organoid versus total cells. Each dot represents an organoid. White dots represent untreated organoids, and blue and orange dots represent organoids treated with 2 μM and 64 μM cisplatin, respectively. The median percentages of A50-EGFP cells for each condition are marked by the triangles at bottom. A total of 2772 organoids with 57,102 cells across three replicate wells for each condition were analyzed for figures **f** and **g**. Source data for **a**–**g** are provided as source data files.

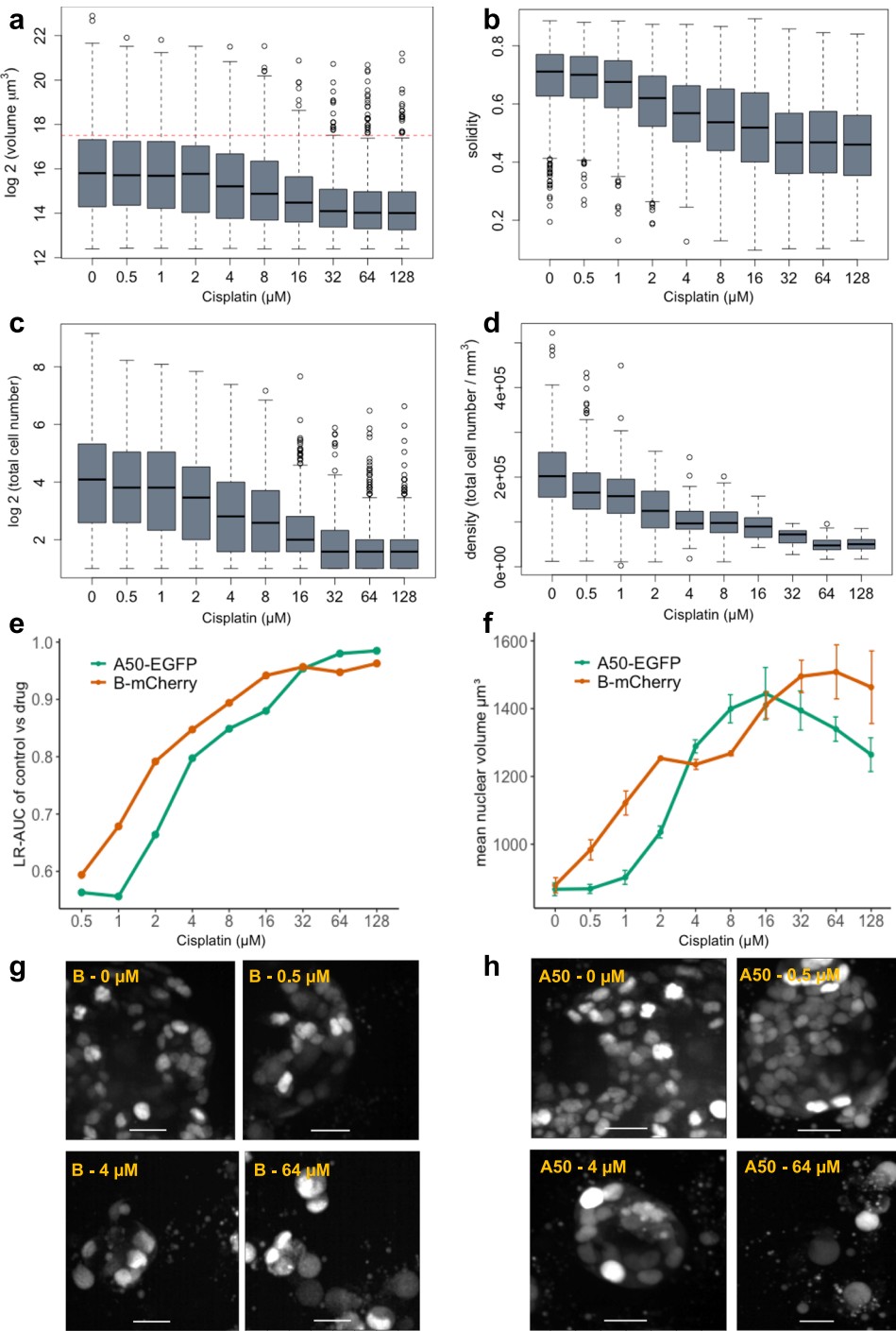

**Fig. 6 | Changes in organoid and nuclear morphological features after cisplatin treatment. a** Organoid volume (μm³). **b** Solidity and **c** Cell number after cisplatin treatment for 4 days is shown. Each experimental condition had three replicate wells. A total of 8382 organoids were analyzed. Horizontal line in the boxplot indicates the median, the box denotes the IQR, the whiskers beyond the box extend to a maximum of 1.5 times the IQR and outliers are shown as dots. Red dotted line on **a** marks the cut-off used to define large organoids (organoid volume > $1.85 \times 10^5$ μm³). **d** Cell density of large organoids ($n = 1077$) after 4 days of cisplatin exposure. Boxplot features same as that described above. **e** Logistic regression classifications based on nuclear morphology, for classification of control nuclei vs. cisplatin-treated nuclei. A50-EGFP and B-mCherry cells are analyzed

separately. LR-AUC indicates area under the curve of the logistic regression classifier for the specified comparison. A total of 352,939 nuclei were used for training and 88,243 nuclei were used for testing. **f** Nuclear volume of A50-EGFP and B-mCherry cells after exposure to a range of cisplatin concentrations. A total of 137,765 nuclei were examined and mean values with standard deviation for triplicate wells are plotted. Source data for **a–f** are provided as source data files. **g** Representative images of B-mCherry cell nuclei at day 4 post cisplatin exposure. A total of 51,269 nuclei were examined. **h** A50-EGFP cell nuclei at day 4 post selected concentrations of cisplatin exposure. A total of 86,496 nuclei were examined. Scale bar represents 25 μm.

could detect quantitative differences in morphology among organoids after cisplatin treatment. Multiple morphological features (volume, volume filled, volume convex, volume bbox, intensity mean, intensity max, intensity min, eccentricity, solidity, Euler number, and inertia tensor eigvals) for every organoid were computed using Cellos. Several features exhibited clear changes post treatment. We observed monotonic decreases in organoid volume (Fig. 6a), solidity as a measure of cell packing (Fig. 6b), and total cell number per organoid (Fig. 6c) as drug concentration increased. These factors showed statistically significant differences ($p < 0.05$) compared to control even at low cisplatin thresholds, e.g. solidity declined at 1 μM cisplatin, $p = 4.13e{-}10$, cell number also decreased at 1 μM Cisplatin, $p = 5.87e{-}05$, and organoid volume was reduced at 4 μM cisplatin, $p = 4.44e{-}08$.

Unexpectedly, despite the decrease in average volume at high cisplatin concentrations, there were still a number of remnant large organoids. 2.5% of organoids in the 64 and 128 μM cisplatin wells had volume greater than $1.85 \times 10^5 \, \mu m^3$ (red dotted line in Fig. 6a), the threshold for the largest quartile of untreated organoids. Such large organoids ($>1.85 \times 10^5 \, \mu m^3$) exhibited different morphological characteristics in the untreated versus treated wells. For example, large organoids showed 4.25-fold lower cell density in the 64 μM wells compared to untreated organoids (Fig. 6d). The treated large organoids also showed lower solidity, lower eccentricity (minor/major axis), and higher Euler number (measure of holes present) (Supplementary Fig. 11a–d). This suggests that the structure of the large organoids appears to be maintained despite a reduction in cellular content (as measured by lower cell density, solidity, and higher Euler number) and an increase in debris arising from dead cells (Supplementary Fig. 12). Thus, Cellos is able to quantify distinctive morphological changes at the individual organoid level as the result of escalating doses of chemotherapy.

We next studied whether Cellos could classify response based on nuclear morphology changes caused by cisplatin treatment. To do this, we trained binary logistic regression models to classify untreated cells versus treated cells for each concentration of cisplatin treatment. A total of 235,957 nuclei were used for this analysis. Separate sets of classifiers were trained for the A50-EGFP and B-mCherry clones. The inputs to the classifiers were Cellos-identified morphological features (volume, volume filled, volume convex, volume bbox, intensity mean, intensity max, intensity min, eccentricity, solidity, Euler number, and inertia tensor eigvals) for every nucleus. For each model, 80% of the dataset was used for training and the remaining 20% was used for testing. In addition, area under the ROC Curve (AUC) was used to evaluate the model. As expected, the logistic regression classifiers showed increasing AUC for cells exposed to higher cisplatin concentrations for both clones (Fig. 6e). However, even at low drug concentrations, the two clones, A50 and B already started to show morphological differences with -0.6 AUC at 1 μM, to 0.7 AUC at 2 μM cisplatin. At higher cisplatin concentrations (>4 μM), the classifier gave AUCs of >0.8 and at the highest dose of cisplatin (128 μM) the AUC was >0.95 for the classification of untreated versus cisplatin treated A50 and B nuclei. This suggests that 3D nuclear morphology can be used as a quantitative surrogate for chemotherapeutic effect.

To verify that these results were not fluorescent label-specific, we repeated the classifier analyses using the reversed fluorescent-labelled nuclei (B-EGFP and A50-mCherry). AUCs were robust despite label flipping (Supplementary Fig. 13a, b), with the maximum AUC difference across the flips being only 0.0677, observed at 64 μM for B-mCherry vs. B-EGFP. This confirms that when a cell is exposed to cisplatin, changes in their nuclear morphology occur in a dose dependent manner, and that these changes are sufficient to build discriminative models regardless of the fluorescence marker used.

Nuclear volume and fluorescence intensity were the morphological features that provided the greatest contribution in the logistic regressions. Thus, we further analyzed these features individually.

Fluorescence intensity decreased with cisplatin concentration for both clones (Supplementary Fig. 13c), indicating that the cellular stress induced by drug exposure might reduce either the label's transcription or translation. Nuclear volume exhibited a more complex behavior, initially increasing with cisplatin concentration for both A50 and B (Fig. 6f), which was also apparent by visual inspection (B clone: Fig. 6g, A50 clone: Fig. 6h). Interestingly, the B clone showed a faster increase in nuclear volume compared to the A50 clone at lower doses of cisplatin, consistent with the logistic regression AUC values and with the greater sensitivity of B clones to cisplatin (Fig. 6e). However, A50 cells showed a greater decline in nuclear volume than the B clone at concentrations >32 μM cisplatin. Decreases in nuclear sizes have been observed in cells undergoing apoptosis[28,29]. Thus, the early reduction in A50 nuclei volumes could indicate the initiation of apoptotic pathways which would correlate with the increased cell death in A50 clones at high cisplatin concentrations compared to the B clone[30]. This confirms that B cells survive better at high drug concentrations, but Cellos also provides insightful information that these surviving cells are not normal and show cellular stress via larger nuclear volume. While the precise meaning of these morphological changes is not clear, the ability to discern morphological alterations on a single cell level can motivate new experiments, especially around the biology of cellular responses to different levels of drug exposure. Our observations suggest that the nuclear effects at cytotoxic doses <32 μM cisplatin are different from the nuclear effects at doses >32 μM.

## Cellos reveals spatial relationships between cells in organoids

3D segmentation at cellular resolution allows for analysis of cell spatial relationships, which may reveal ecological interactions among clones. To interrogate such interactions, we analyzed how clones were organized within heterogeneously mixed TNBC organoids. For each clone in each organoid, we calculated the *localization score*, which quantifies how often a cell's adjacent cells are of the same clonal type, within a specified cell number window size. This score is normalized for the clonal fraction in the organoid. The higher the localization score, the higher the co-localization of cells of the specified clone with themselves (see Methods).

We observed that cells of a given clone tend to co-occur with others of the same clone, as demonstrated by a decrease in localization score with increasing window size. This was true both in the presence and absence of cisplatin treatment, and for both clones B (Fig. 7a) and A50 (Supplementary Fig. 14a). These results indicate that clones form small spatial clusters within the organoids. Co-localization was stronger in the organoids having only a small fraction of one clone, whether the minor population clone was B (Fig. 7b) or A50 (Supplementary Fig. 14b). This effect was consistent even in the 10-cell windows (Supplementary Fig. 14c, d). An example of a heterogeneously mixed organoid (consisting of 20% B clone fraction) with strong clone localization (localization score = 3.1) is shown in Fig. 7c. In organoids with equal fractions of the two clones, we observed cases both where clones are well mixed (Fig. 7d, localization score = 1.25) and where they form separate clusters within the organoid (Fig. 7e, localization score = 1.53).

Local cell division could generate spatial clusters of cells within organoids, but they could also arise from ecological affinity between cells of the same clone. Comparing homogenously mixed organoids to heterogeneously mixed organoids allows one to distinguish whether spatial clusters arise from cell division or ecological affinity. This is because homogeneously mixed organoids should demonstrate the cell division effect but not the ecological affinity effect. We observed significantly higher clone localization in heterogeneously mixed organoids compared to homogeneously mixed organoids (Fig. 7f, g). This was the case for comparison of homogeneous B organoids to B-A50 mixed organoids ($p = 1.5e{-}9$), as well as for comparison of homogeneous A50 organoids to A50-B mixed organoids ($p = 5.2e{-}6$). Thus, these clones have greater ecological affinity for cells of their same type.

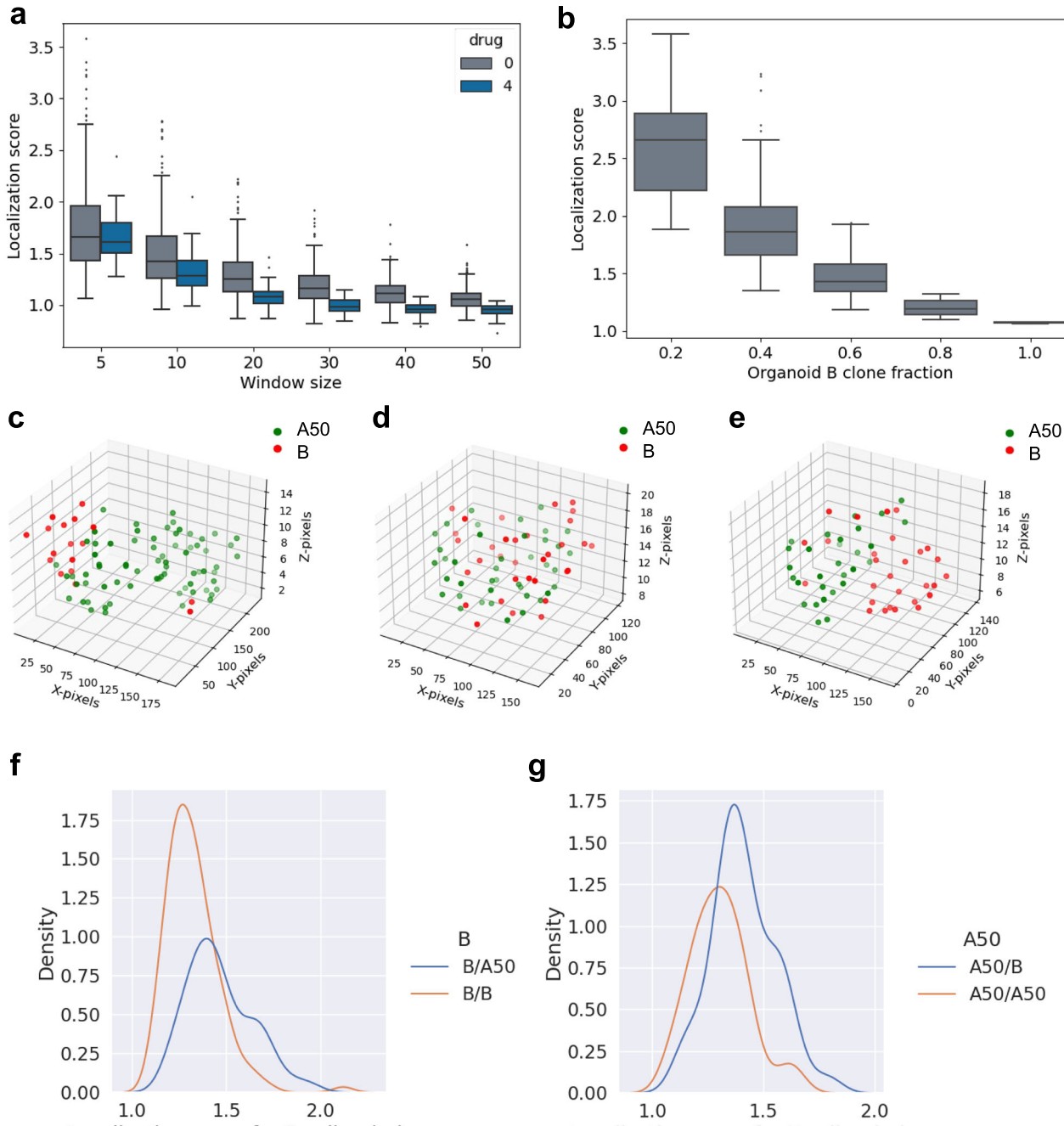

**Fig. 7 | Investigation of cell-cell spatial relationships within organoids.**
**a** Localization score vs. window size (5, 10, 20, 30, 40 and 50 cells) for B-mCherry cells within heterogeneously mixed organoids. A total of 271 organoids from control or 4 μM cisplatin treated conditions are shown. **b** Localization score for B-mCherry cells in organoids (5 cell window), stratified by B clone fraction in the organoid. Organoids ($n = 232$) are binned in increments of 0.2 for the B cell fraction. For boxplots in **a** and **b**, the median is shown by the horizontal line in the boxplot, the box denotes the IQR, the whiskers extend to a maximum of 1.5 times the IQR and outliers are shown as dots. **c–e** 3D spatial locations of nuclei in representative individual organoids showing different patterns of colocalization. The 3D scatter plots for each of the three representative organoids consists of 88 (left panel), 74 (center panel) and 57 (right panel) nuclei respectively. **f** Distribution of localization score of B clones (in 5-cell window) when mixed with A50 (blue line) or with alternately labeled B (orange line). Kernal density estimate is used to visualize distribution of the data consisting of 212 organoids. **g** Distribution of localization score of A50 clones when mixed with B (blue line) or with alternately labeled A50 (orange line). Kernal density estimate was applied on 152 organoids. Source data for **a–g** are provided as source data files.

We also used radial distribution as an alternate metric to assess the proximity of A50 cells to either B or other A50 cells (see methods). Radial distribution analysis quantifies the frequency of cells of each clonal type to be near cells of the same type within a specified radius. The higher the radial localization score, the higher the co-localization of cells of the specified clone. In such an analysis, we observed significantly higher clone localization in heterogeneously mixed organoids compared to homogeneously mixed organoids (Supplementary Fig. 14e, f). This was the case when comparing homogeneous B organoids to heterogeneous B-A50 organoids ($p = 7.673e-5$) (Supplementary Fig. 14e), as well as for comparison of homogeneous A50 organoids to heterogeneous A50-B organoids ($p = 2.512e-5$) (Supplementary Fig. 14f). Note that to avoid the effect of clonal fraction on localization score, we restricted these analyses to organoids with

comparable clonal fractions, with both clones within a range of 0.4-0.6.

## Discussion

We have presented Cellos, a high-throughput pipeline for segmenting and analyzing organoids at the cellular level. The distinctiveness of Cellos compared to other organoid segmentation methods is that it can segment large numbers of organoids at cellular resolution within each well, in flexible culture conditions, and in true 3D (Supplementary Table. 2). These aspects provide accurate high-throughput quantification with little need for user-specified parameter tuning. Further advantages of Cellos include its abilities to accurately segment organoids and cells from different imaging modalities; to robustly determine cell ratios in wells; to generate $IC_{50}$ curves based on cell segmentations within images; and to identify organoid and nuclear morphological features associated with treatment response. Moreover, Cellos enables 3D analysis of cell spatial relationships, which are valuable to the assessment of tumor microenvironmental interactions.

The high-throughput capacity of Cellos is crucial to its value for drug or gene knockout screening analysis. To clarify its scalability, ~100,000 organoids and ~2.35 million cells were analyzed for the experiments presented here. In addition, the analysis of multiple wells was parallelizable, with the time for 60 wells processed by 60 CPUs being similar to the time for one well with one CPU. Imaging speed and size of the data in 3D are also important practical considerations for large scale image-based assays. To minimize imaging time, we used coarser z-sampling intervals of 5 μm while still achieving accurate segmentation, requiring far less data. Prior studies have used z-stack intervals of 0.122 μm to study *Caenorhabditis elegans*[31] and 1.2 μm intervals for cleared spheroids[32]. Axial sampling of 5 μm was chosen because our initial studies indicated that resolutions finer than 5 μm did not appreciably improve segmentation accuracy. For the majority of experiments we limited the number of z-slices to 100 and imaged the same z-range for all wells in a plate, which restricted identification of organoids to certain regions of the well. Because organoid distribution within wells can be variable, this led to differences in cell counts across replicate wells. To address this problem, Cellos includes a customized method for distinguishing image areas that contain organoids in focus within the acquired image of every well (Supplementary Fig. 15a–c). This area estimate can then be utilized to determine organoid and cell densities, allowing for a more robust comparison between wells. In addition, Cellos can generate quantifications based on clonal ratios within a well, which are even less sensitive to this issue since ratios calibrate for absolute cell numbers. To decrease storage, images generated by Cellos were saved as zarr arrays[33], which required ~12GB to store an image of one well (with a size of $3 \times 101 \times 5080 \times 5080$ pixels). Altogether, when pre-processing and morphological feature characterization are included, it took on average ~1.9 h with CPU efficiency of 91.2% and ~100 GB of computational memory to segment 550 organoids, and ~1.4 h with CPU efficiency of 82.31% and 6.86 GB of computational memory to segment 4837 nuclei. These computations are straightforward to execute because the pipeline has been optimized to work on high performance computing systems.

A major challenge in cancer therapeutics is deciphering the effects of multicellular interactions. For example, the tumor microenvironment contains diverse tumor clones, immune cells, endothelial cells, and tumor-associated fibroblasts, which may each provide new target pathways for drug development[34–36]. Non-organoid approaches to evaluate how drugs interact with the tumor microenvironment have included measurements in animal models, tissue explants, or observations in clinical samples[37], all of which are challenging to scale. Clonal heterogeneity in tumors has also long been studied through 2D techniques such as immunohistochemistry. In contrast, in vitro 3D organoid technologies allow interrogation of drug and cellular

combinatorics based on high numbers of independent multicellular structures. Cellos facilitates interpretation of these parallel measurements in organoids by individually quantifying and analyzing organoids and their cells in 3D to elucidate growth and pharmacology effects.

Cellos is effective in 3D cell segmentation despite complexities such as the difficulty of identifying cell boundaries in packed organoids, as well as inherent differences in cellular morphology across cell types, which can be changed further by drug treatment. We genetically engineered the expression of fluorescent tagging in many of our experiments to provide controlled markers for distinguishing the behavior of different cancer clones. However, we also observed that nuclear dyes such as Hoechst can be effectively used for segmenting nuclei by Cellos. Multiplex analysis of cellular morphology is likely to become increasingly common in the field, and we expect that expansions of Cellos techniques to viability dyes and other markers will be beneficial for assessing responses of organoids and cells to different drug concentrations.

In our analysis of the pharmacological dynamics of TNBC organoids with complex multiclonal cisplatin sensitivity, we could accurately recapitulate the $IC_{50}$ curves of well-based luminescence assays using Cellos including subtle non-monotonic changes in relative drug sensitivity. Moreover, the morphological characterizations enabled advanced associations that have not been previously possible. For example, we were able to quantify how nuclear volume changes with cisplatin, how volume tracks with viability changes observed in $IC_{50}$ curves, and the population dynamics of organoids after drug exposure. While we have focused on intuitive features such as volume and fluorescence intensity and their contribution to logistic classifiers, more general image analysis algorithms, such as convolutional neural networks[38,39], may provide even greater potential for classification of the biological states of organoids and cells.

Another promising direction enabled by Cellos is the quantification of cell-cell spatial relationships−an inherently three-dimensional problem. We were able to distinguish the affinity of a cancer cell clone to localize with other cells of the same clone from the effects of local cell division. An interesting future direction would be to study if such affinities can be altered to modulate treatment response. In any case, the detection and quantification of ecological affinities in model systems will be valuable for understanding the interactions within tissue microenvironments. Such analyses can clarify the impacts of cell juxtapositions resulting from processes including cell division, motility, cytokine signaling, and cell-cell ligand-receptor interactions.

In conclusion, we report a high-throughput pipeline for true 3D volumetric organoid and nuclei segmentation, enabling quantitative evaluation of cell counts, spatial relationships, and nuclear as well as organoid morphology affected by treatment. Cellos opens new opportunities for organoid experimentation and the elucidation of multicellular phenotypes from imaging.

## Methods

### Clonal line establishment and nuclear fluorescent labelling

The TM00099 PDX model have previously been established under the protocols approved by The Jackson Laboratory IRB (Protocol #121200011)[18]. In this study, primary cell cultures were derived from TM00099 PDX tumor fragments. Briefly, tumor fragments were disassociated in 2 mg/ml Collagenase Type IV (Invitrogen) for 1−2 h at 37 °C on a rocker, passed through 100 μm cell strainers and harvested cells were washed and plated on irradiated 3T3-J2 feeder cells and maintained in 37 °C and 7.5% $CO_2$ in culture medium as described by Liu et al.[40] After in vitro expansion of the primary cell cultures, single human cells labelled with Anti-Human HLA-ABC APC (cloneW6/32, eBiosciences, Cat. no. 17-9983-42) at a dilution of 5 μl/5 × 10⁶ cells were isolated using flow cytometry. Forward scatter was plotted against side scatter to gate singlet cells after which APC positive cells were

identified and gated based on unstained control cells. Single cells were sorted and further expanded to establish clonal lines. Two clonal lines defined as A50 and B were then stably transduced with lentivirus expressing nuclear EGFP or mCherry. pTRIP-SFFV-EGFP-NLS (Addgene, Plasmid #86677) was used for the lentiviral nuclear EGFP construct. mCherry sequence was subcloned from pLenti6-H2B-mCherry (Addgene, Plasmid #89766) to replace the EGFP sequence thus generating a pTRIP-SFFV-mCherry-NLS construct.

The two lentiviral plasmids were packaged into lentivirus by transfecting 5 μg of each plasmid with psPAX2 (Addgene, Plasmid #12260) and pMD2.G (Addgene, Plasmid #12259) in HEK293T (ATCC, Cat. no. CRL-3216) cells with Lipofacamine 2000 (Invitrogen, Cat. no. 11668019). Lentivirus collected after transfection was concentrated using 3 P Lenti-X™ Concentrator (Clontech Labs, Cat. no. NC9833735) using manufacturers protocol. Clonal cell lines were infected with the lentivirus and selected for by sorting for nuclear EGFP or mCherry cells respectively. All clonal lines were authenticated by validation of clonal line specific structural variations and routinely tested for Mycoplasma contamination using the MycoAlert PLUS Mycoplasma Detection Kit (Lonza, Cat. no. LT07-710).

### 3D cell culture assays and imaging

For 3D culture assays, fluorescence-activated cell sorting (FACS) was used to isolate EGFP or mCherry cells of the desired clone. Single cells were gated using standard gating on forward scatter vs side scatter plots and cells positive for the flourescent labels were then gated on using cells without any flourescent labels as negative controls. Sorted cells were seeded at a density of 30,000 cells per well in triplicate in 96 well plates (PerkinElmer) coated with 35 μl Matrigel Growth Factor Reduced (BD Biosciences, Cat. no. 356230) on day −3. Cells were cultured in 37 °C and 7.5% $CO_2$ for three days for organoids to form. On day 0, media was replaced, and organoids were treated with 9 doses of cisplatin (Selleck Chemicals, Cat. no. S1166) in two-fold serial dilutions in the range of 0.5 to 128 μM. After 96 hours of drug exposure, on day 4, CellTiter-Glo® 3D Cell Viability Assay (Promega, Cat. no. G9681) was used for luminescence-based assay readouts using manufacturer's protocols.

For image-based readouts, plates were imaged at day 0 or day 4 using the Opera-Phenix High-Content Screening System (PerkinElmer). Twenty-five fields per well, were imaged using a 20× water objective. In each field, 101 z-stacks at 5 μm separation were imaged in three channels−brightfield, EGFP and mCherry. Cells were maintained at 37 °C and 7.5% $CO_2$ during the imaging. For training set experiments, nuclei were counterstained with a final concentration of 1 μg/ml Hoechst 33342 Solution (Thermo Scientific, Cat. no. PI62249) for 30 minutes at 37 °C before imaging.

For generation of heterogeneously mixed organoids, A50-EGFP and B-mCherry cells (experiment-1) or A50-mCherry and B-EGFP (experiment-3) cells were mixed in 50:50 ratios and treated with drug as described above and imaged at day4 post drug exposure. For generation of experiment-2, homogeneously mixed organoids consisting of A50-EGFP and A50-mCherry (or B-EGFP and B-mCherry cells) were generated by mixing them in varying rations of EGFP:mCherry namely −20:80, 40:60, 60:40 or 80:20 respectively. These organoids were imaged at day 0 and day 4. For experiments 4 and 5, homogeneously mixed organoids consisting of equal proportions of A50-EGFP and A50-mCherry (experiment-4) or B-EGFP and B-mCherry cells (experiment-5) were generated and treated with cisplatin and imaged as described above.

### 3D culture and imaging for breast cell lines

TNBC cell line MDA-MB231 was a kind gift from Min Yu's laboratory (University of Maryland) and was cultured in DMEM (Gibco) supplemented with 10% fetal bovine serum (Gibco) and 1% Penicillin/Streptomycin (Sigma). HCC1806 (ATCC, Cat no. CRL-2335) was cultured in DMEM (Gibco) supplemented with 15% Fetal bovine serum (Gibco) and 1% Penicillin/Streptomycin (Sigma). Breast cell line MCF10A (ATCC, Cat. no. CRL-10317) was maintained in DMEM/F12 (Gibco) supplemented with 5% horse serum (Gibco), 1% Penicillin/Streptomycin (Sigma), 20 ng/ml EGF (Peprotech), 0.5 μg/ml hydrocortisone (Sigma), 100 ng/ml cholera toxin (Sigma), and 10 μg/ml insulin (Sigma)[41] and all cell lines were maintained in 5% $CO_2$ at 37 °C. Cells were routinely tested negative for mycoplasma using the MycoAlert Mycoplasma Detection Kit (Lonza) and cell aliquots from early passages were used. For 3D experiments, cells were seeded at varying densities on 35 μl of growth factor reduced Matrigel (BD Biosciences) per well of 96 well plates, allowed to grow for 4–5 days and imaged using the Opera Phenix system. For MDA-MB231, cells were seeded at increasing seeding densities of 2000 (low), 3000 (medium) or 4000 (high) cells per well and imaged after 4 days in 3D culture. HCC1806 and MCF10A cells each were seeded at three seeding densities (low, medium, and high) of 5000, 7000 or 10000 cells per well and 200, 500 or 700 cells per well respectively and imaged after 5 days in 3D culture. MDA-MB231 and HCC1806 organoids were stained with Hoechst (Invitrogen, Cat. no. 62249) at final concentration of 5 μg/ml and Calcein AM (Invitrogen, Cat. no. C3099) at a final concentration of 1 μM for 30 min at 37 °C and imaged using above mentioned imaging conditions for a total of 165 individual z stacks per well. MCF10A organoids were stained with 5 μg/ml Hoechst (Invitrogen) and imaged in a similar manner. For all cell lines and seeding conditions, images were collected from at least three and up to five replicate wells for each cell line and seeding density.

### HCC1806 3D organoids drug screen

For drug experiments on the HCC1806 cell line, 7000 cells were plated on 35 μl of growth factor reduced Matrigel (BD Biosciences) per well of 96 well plates on day −3 and organoids were allowed to form for three days. On day 0, media was replaced, and the cells were treated with two drug concentrations (Dose1 and Dose2) each for three drugs namely, Cisplatin, Doxorubicin and Docetaxal in triplicate conditions. Organoids were treated with 15 μM and 30 μM Cisplatin (Selleck Chemicals), 1 μM and 2 μM Doxorubicin (Selleck Chemicals, Cat. no. S1208) and 6.5 nM and 13 nM Docetaxel (Selleck Chemicals, Cat. no. S1148) respectively. Organoids were maintained with drugs for 72 h, stained with 5 μg/ml Hoechst (Invitrogen), 1 μM Calcein AM (Invitrogen) and 1 μM DRAQ7 (Abcam, Cat. no. ab109202), incubated for 30 min at 37 °C and imaged using above mentioned imaging conditions for a total of 200 individual zs per well. CellTiter-Glo® 3D Cell Viability Assay (Promega) was then used for luminescence-based assay readouts using manufacturer's protocols.

### Analysis protocol overview

We developed a customized pipeline combining python and shell scripts to analyze 3D organoids both at the organoid and cellular level. The customized pipeline is written to process an entire multiwell plate. Multiple wells can be processed at the same time on a high-performance computing cluster by taking advantage of multiple central processing unit (CPU) cores. The main steps for the pipeline are: 1. Exporting and organizing image data, 2. 3D segmentation of individual organoids, 3. Cropping individual organoids and 3D segmentation of nuclei in each organoid, 4. Computing morphological features and saving output information, 5. Calculating area of imaged well with organoids, 6. Post segmentation analysis such as IC50 estimates and cell co-localization analysis. The pipeline code is available on GitHub.

### Exporting and organizing image data

The image data were exported from the Opera Phenix high content screening confocal microscope using the Harmony High-Content Imaging and Analysis Software (PerkinElmer). The resulting folder

contains subfolders with tiff files (Images) and xml file (metadata). Each tiff file is a single image from one well, one field, one x,y-plane and one channel. This means for one well we had 7500 tiff files (25 fields × 101 planes × 3 channels). We developed an automatic protocol that organized all tiff files from one well and saved them as zarr arrays. To do this, first we created an empty zarr array with size equal to a whole well image (the size of the image is collected directly from xml file). Then we put individual planes and channels from the same field together in the zarr array. Lastly, we stitched all the fields of a well together.

## 3D organoid segmentation

Organoid segmentation had two major steps, preprocessing of the image and segmentation of the organoids. Most of the algorithms used were from python scikit image processing package[20]. The preprocessing step reduces debris and intensity differences to increase accuracy of organoid segmentation. To do this: 1. The image is converted to gray scale, and a threshold is used to create a binary image, 2. Binary opening and dilation algorithms are then used to close small holes and remove small dots. For simplicity this is done on the 2D "max-projection" image, corresponding to the projection of the image maximum among all z-stacks. 3. The binary image is then multiplied by the original image to produce a "cleaned" image. For the organoid segmentation step: 1. The "cleaned" image is converted into gray scale, and a multidimensional gaussian filter is used to remove noise. 2. The threshold triangle method is then used to generate a binary image. Note: All the above steps are done on single field instead of an entire well image because fields often vary in noise and intensity. 3. The individual fields are stitched together. Objects in the stitched image are given a label using the "measure label" algorithm, 4. Unwanted small objects are removed using the "morphology remove small objects" algorithm. "Measure regionprops table" algorithm is used to generate a table of measurements for each organoid containing: label, centroid, 3D bounding box (bbox), volume, volume (bbox, filled, and convex), major and minor axis length, euler number, extent, intensity (max, min, and mean), and inertia tensor eigvals. 5. The output is stored in a csv file with the file name containing the row and column number of that well.

## StarDist-3D training

To annotate ground truth images for training a StarDist-3D model, we followed guidelines from Stardist[42] GitHub page. Ground truth images were annotated manually using Labkit[43] a plugin in Fiji. Every nucleus was given a label. In total 36 organoids from different wells and biological conditions were labeled. 10, 12 and 14 organoids had cells which were fluorescently labeled with Hoechst, EGFP and mCherry respectively. Hoechst labeling was performed to diversify the training set. Among these 36 organoids, 24 had been treated with cisplatin. In total 3,862 nuclei were annotated. The data was augmented using elastic deformation, noise/intensity shift, and flip/90-degree rotation. Note that we did not see significant changes in validation metrics when in total 30 organoids were used (precision = 88.15 ± 4.7, recall = 87.47 ± 3.76 and accuracy = 78.31 ± 6.09). For training, we used parameters based on an example from the StarDist GitHub. The model configuration was: anisotropy = (9.0, 1.0, 1.0), backbone = ResNet, number of rays = 64, patch size = (16, 128, 128), epochs = 400. The median object size was (2, 18, 18). Thus, a network view of (17, 30, 30) was used to make sure at least one nucleus is seen by the network. "KFold scikit learn model selection" was used to split data into six folds for cross validation, with 30 and 6 organoids for training and validation, respectively. The trained model was then used to segment nuclei in organoids. It is important to note that non-star-convex-shaped nuclei cannot be segmented properly by the model.

## 3D nuclei segmentation

The segmentation of nuclei in organoids was done using the trained model. 3D bbox information of each organoid was used to extract each organoid in the saved zarr image of cells. To segment nuclei in the organoid: The organoid image was normalized. In detail, each individual channel of each organoid is normalized before the nuclei segmentation step. This normalization is percentile based and is part of the StartDist method[44]. The trained model was used to segment individual nuclei, by giving each nucleus in the organoid a label. The "scikit image measure regionprops table" algorithm was used to generate a table of measurements for each nucleus: label, centroid, 3D bounding box (bbox), volume, volume (bbox, filled, and convex), major and minor axis length, euler number, extent, intensity (max, min, and mean), inertia tensor eigvals, and eccentricity (which was calculated as minor axis length/major axis length). Each fluorescence channel was processed and segmented separately. The data for all nuclei in a well were then stored as a csv file (one csv file per well) with the file name contains the row and column number of that well. It is important to note that Cellos segments nuclei regardless of their fluorescence intensity (supplementary Fig. 2d). The post-segmentation analysis used the csv files generated in the organoid and nuclei segmentation steps.

## Calculating area of imaged well with organoids

Cellos includes a customized method for distinguishing image areas that contain organoids in focus within the acquired image of every well (Supplementary Fig. 15a–c). To do this, the whole well image with segmented organoids is used as input. To calculate the area all the processes are done on the 2D "max-projection" image. First, any very big objects that are not organoids are removed (note: we rarely have these objects), then any space between adjacent organoids is closed using "scikit image dilation and binary closing". Second, the non-zero pixels (pixels with organoids) are counted using "Opencv-python (cv2) countNonZero" package. Finally, the number of pixels is converted into desired unit such as $mm^2$, based on the size of an image pixel in μm. Notably, this step to calculate the area is done during organoid segmentation and can be skipped if not needed.

## Detecting and removing dead nuclei from analysis

Cellos includes a customized step to detect dead nuclei and remove them from the subsequent analysis. To do this, for each individual cropped organoid, we first normalize its dead cell dye (DRAQ7) channel intensity so that the intensity is between 0-1. This is performed because intensity values may vary across distinct organoids. We then proceed to evaluate each segmented nucleus within an organoid. We calculate the average intensity of the DRAQ7 channel in the segmented nucleus bounding box region. The resulting calculated number is then used to categorize each nucleus as live or dead. To make this classification we rely on a predetermined threshold of nucleus with intensity >0.1 as dead. The threshold was chosen by evaluating several histograms illustrating the average DRAQ7 intensity of all the nuclei in several different wells.

## IC$_{50}$ estimation

Cell density for each well was calculated by dividing the number of segmented cells by the customized calculated area of the imaged well with organoids. Cell densities for each drug-treated well were compared to cell densities from control wells (medium only). IC$_{50}$ values were estimated using R package *nplr (*N-Parameter Logistic Regression. R package). For the luminescence based IC$_{50}$ assay, luminescence values were compared to control wells and IC$_{50}$ values were calculated as mentioned above. Error bars represent the standard deviation of triplicates for each condition.

## Cells co-localization analysis

The co-localization analysis was performed on organoids consisting of two cell populations each labeled with either EGFP or mCherry. To calculate co-localization score of a specific fluorescently labeled clone, for example B-mCherry in an organoid: 1. Centroid of nucleus was computed for each cell in the organoid. 2. All pairwise distances (Euclidean distance) between nuclei centers were computed. 3. For each B-mCherry cell, the neighboring cells were ranked by distance in the window of $K = 5$, 10, 20, 30, 40, and 50 cells. 4. B-mCherry cells proportion was then calculated in each window ($K$) and normalized for the overall proportion of B-mCherry cells in the organoid. This was performed to avoid the bias of varying proportions of B-mCherry cells across all organoids. 5. Co-localization score for B-mCherry in the organoid for a given $K$ was then calculated by averaging the co-localization score of all B-mCherry cells in the organoid. We hereafter denote the co-localization score for clone type $X$ in an organoid, $Lo$ as,

$$Lo = \frac{1}{n}\sum_{i=1}^{n} Lc_i \tag{1}$$

where, $n$ is the total number of cells of clone type $X$ in the organoid and $Lc$ is the co-localization score of a cell of clone type $X$ in a given $K$ and is defined as,

$$Lc = \frac{X_n}{K * R_x} \tag{2}$$

where, $X_n$ is the number of cells of clone $X$ in given $K$, and $R_x$ is the proportion of clone $X$ in the organoid.

## Radial localization score calculation

To calculate the radial localization score of a fluorescently labeled clone, for example B-mCherry in an organoid: 1. The nuclear centroid was computed for each cell in the organoid. 2. All pairwise distances (Euclidean distance) between nuclei centers were computed. 3. The longest distance in the organoid was noted ($r_{max}$). 4. For each B-mCherry cell, the cells between $[0, r_i]$ were identified, where $r_i$ was calculated for 6 equally spaced values from $r_{max}/6$ to $r_{max}$. 5. The proportion of B-mCherry cells was then calculated for each $r_i$ and normalized by the proportion of B-mCherry cells in the organoid. The subsequent steps are the same as for the window-based co-localization analysis. Note for Supplementary Fig. 14e, f, results are shown at $r_i = 2(r_{max}/6)$. Results at other values for $r_i$ are qualitatively similar.

## Statistics and reproducibility

All experiments were designed to have at least three replicate wells for each condition. This was determined based on feasibility of imaging within a reasonable time frame given the scale of imaging required for each assay (100 z stacks × 25 fields × 4 fluorescent channels) where each field is 1080 × 1080 pixels in dimension. All samples (organoids and nuclei) available in acquired images of good quality or met our thresholds as described in manuscript were included in the analysis. In instances where image quality was poor (for example, organoids were not captured in acquired image), the replicate was disregarded from the analysis, and this was noted in the respective figure legends. Sample sizes were determined to be sufficient based on the magnitude of data analyzed in different conditions and consistency observed across replicates. Experiments were not randomized, and the investigators were not blinded to allocation during experiments and outcome assessment. All additional statistics performed are described in the paper. All t-tests were two-tailed. All error bars indicate standard deviation or 95% confidence interval and are described in figure legends. In all boxplots, the center line represents the median, the box limits represent the upper and lower quartiles, and the whiskers represent the 1.5× interquartile range.

## Reporting summary

Further information on research design is available in the Nature Portfolio Reporting Summary linked to this article.

## Data availability

Representative dataset of raw images for one well and its metadata generated in this study has been deposited in the figshare database under (https://figshare.com/articles/dataset/cellos_data_zip/21992234)[42]. Manually annotated dataset used to train the nuclei segmentation model have been deposited in GitHub under the Zenodo accession code[45]. Considering the volume of imaging data generated in this study, the remaining data are available in two weeks upon request. All source data are provided with this paper. Cell lines will be made available in four weeks upon request. Source data are provided with this paper.

## Code availability

Cellos pipeline code has been deposited in GitHub under the Zenodo accession code[45].

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

## Acknowledgements

The authors gratefully acknowledge the contribution of the Flow Cytometry service, Single Cell Biology service, Microscopy core service, Cyberinfrastructure high performance computing resources, and Research IT at The Jackson Laboratory for expert assistance with the work described in this publication. These shared services are supported in part by the JAX Cancer Center (P30 CA034196). The authors would like to thank Ali Foroughi Pour for helpful comments. The authors also would also like to thank Jim Peterson, Yi Li, Philipp Henrich, and Erick Ratamero for helpful discussions about organoids and nuclei segmentation, Peter Sobolewski and Fernando Cervantes for helpful guidance on generating 3D image representations. Research reported in this publication was supported by NIH grant R01CA230031, JC and NIH/NCI Cancer Center Support Grant P30 CA034196.

## Author contributions

P.M. designed the study, generated training dataset, developed the software, performed data analysis, wrote the paper, and revised the paper. P.K. designed the study, generated training dataset, performed data analysis, generated experimental data, wrote the paper, and revised the paper. D.M. contributed to software development and reviewed the paper. S.N. performed data analysis and reviewed the paper. J.N. contributed to design of the study and reviewed the paper. MB contributed to generating the breast cell line organoids, E.C. contributed to imaging setup and O.A. contributed to experimental design and reviewed the paper. E.L. designed the study, interpreted data, wrote the paper, and supervised the project. J.C. designed the study, interpreted data, wrote the paper, and supervised the project.

## Competing interests

The authors declare no competing interests.
