## [Peer Review File · Nature Communications]

Reviewers' comments:

Reviewer #1 (Remarks to the Author)

The work by Mukashyaka, P., and team entitled “Cellos: High-throughput deconvolution of 3D organoid dynamics at cellular resolution for cancer pharmacology” is a showcase of a novel method to segment and analyze 3D models at cellular resolution with high throughput that is much required in the field of drug screening research, especially cancer drug screening.

The novel image analysis pipeline showed the accuracy and flexibility that can be used to deal with research questions at the organoids and the nuclei levels. The authors started by describing the TNBC model organoids with drug sensitivity heterogeneity (made from the PDX model TM00099) and derived and applied this cancer line to a few models to test the capability of the analysis pipeline.

Overall, this work demonstrated the advancement of the methods used for organoid-based drug screening, which is still inadequate and may be of interest to people who work in drug development and cancer modeling, etc.

However, there are some issues that may need further clarification.

1. Has the pipeline been validated and tested in the other 3D cancer models, with different morphology such as the organoids with cyst-like, dispersed, compacted, or others? In order to generalize the application of this pipeline, data from 3 or more types of organoids should be included.
2. Interesting finding from the study is the ability of Cellos to reveal spatial relationships between cells in organoids. I strongly recommend that the finding be validated by other independent means.

Reviewer #2 (Remarks to the Author)

The authors have developed a new computational method, called Cellos (Cell and Organoid Segmentation), for high-throughput segmentation and analysis of 3D organoids and their nuclei. The method uses classical algorithms to segment organoids in 3D and a Stardist-3D convolutional neural network to segment nuclei, allowing for analysis of cell densities, clonal population frequencies, and potential cell-cell interactions. Cellos was able to accurately distinguish ratios of distinct fluorescently labeled cell populations in organoids and recapitulate traditional luminescence-based drug response quantifications. The method was tested on triple negative breast cancer organoids containing clonal mixtures with complex cisplatin sensitivities, and was able to identify organoid and nuclear morphology feature changes associated with treatment. Cellos provides powerful tools for high-throughput analysis for pharmacological testing and biological investigation of organoids based on 3D imaging.

Comments.

Several methods for segmenting organoids already exist, such as using automated segmentation methods like watershed segmentation, active contour models, and machine learning-based approaches. What is the advantage of Cellos compare to other existing methods? The authors should describe these points to prove the novelty of Cellos pipeline.

Figure 1A: the relative viability of untreated B- and A50-clones is different (~ 1.1 in B-clone and ~ 0.9 in A50-clone) at the starting point. What is the value normalized to? Is cell proliferation different between these clones? A slow-growing phenotype can affect the drug sensitivity. The author should characterize the proliferation ability of A50- and B-clone cells not only for the pure organoids but also homogeneously or heterogeneously mixed ones.

Figure 1B: scale bars are missing.

Line 90: What 'primary cell lines' means? 'primary' and 'cell line' are contradictory. Do the authors use multiple cell lines, or is it 'a cell line' not 'cell lines'?

The authors show the high accuracy of Cellos nuclei segmentation by the use of only one external dataset (breast carcinoma from Boutin et al.; 184). The authors should investigate if the results can be reproduced in another available dataset.

Can Cellos discriminate between dead cells, quiescent or dormant cells, senescence cells, or proliferating cells by detecting the nuclear fluorescence labeling after drug exposure?

The authors should apply the Cellos method to quantify the treatment response to other patient-derived tumor organoids. Moreover, the degree of cell damage or death induced by the drugs varies depending on the type of drug. Therefore, other drug treatments should be evaluated in addition to cisplatin.

In the cisplatin treated case, damaged nuclei would be much harder to be segmented as the deformity and intensity of the fluorescence substantially varies. Is there any threshold to take the weak intensity of fluorescence into account as nuclei? The images of cisplatin-treated A-50 cell should be shown in Figure 5 not just of the B-mCherry since A-50 would be much more damaged by cisplatin in higher doses.

The authors should show nuclei segmentation images for cisplatin treated case like in Figure 2. More visible pictures should be provided as supplementary information. The pictures in Figure 2a are too small to judge the performance of Cellos.

Line 201: What day0 means? Immediately after making mixture of cell suspensions? Several hours after waiting for spheroid formation? It takes time for the suspended cells to aggregate and further form spheroids. The authors should show the method of making chimeric spheroids more in detail. Time course study of spheroid formation from cell suspension should be performed. Please show the day0 images in Figure 3.

Figure 5: g and f are mislabeled.

Spatial relationship in Figure 6 is intriguing and provides a new biological insight, although dividing the possible cause into 2 mechanisms, cell division and ecological affinity, is too simplified. It is possible that the cells attaching to the similar cells are more advantageous for cell division. Do the heterogeneously mixed cells move in the spheroids to attach to the similar cells each other? Time lapse study would provide useful information. Can Cellos be applied to such analysis? The nuclei segmentation would be useful for the precise tracking of the nuclei.

The potential limitations of this approach, and how they might affect the applicability should be discussed. Please discuss the following points:

- a) Are there any potential errors or inaccuracies in nuclei segmentation due to the high density of cells in organoids?
- b) What are the potential limitations of using a training dataset of only 36 organoids to develop a CNN for nuclei segmentation?
- c) Are there any potential biases in using specific fluorescent labels (EGFP, mCherry or Hoechst) for nuclei segmentation in this study?
- d) While the pipeline is described as high-throughput, what are the potential limitations or bottlenecks? How long actually does it take to analyze large datasets of organoids?

Reviewer #3 (Remarks to the Author)

Synopsis. The authors fine-tuned a single cell segmentation method for organoid/spheroid cultures using a deep learning-based method (Stardist/RESNET). The method/model, referred to as Cellos, was

trained on a curated dataset and tested on a larger library of spheroid images generated elsewhere. Conventional feature extraction was performed and used for downstream analysis. Use applications included quantification of clonal architecture and response to drugs, as well as identification of features that were used to classify genotoxic drug mechanism of action. The methods and experimental approaches implemented here were done with rigor and followed industry standards. Example data and analytical scripts are made available in a well-documented/structured GitHub location that supplements the text. My only concerns are that the author should place more emphasis on what truly differentiates their approach from previous generations of analysis.

Itemized list of concerns:

On line 150, manual quantification was performed as a benchmark. Was only one “expert” used or did multiple experts quantify the data to evaluate operator error?

LabKit, a more conventional ML-segmentation approach, was used to be used “ground truth” to benchmark Cellos. The data presented shows a generally high IoU between the methods, indicative of comparable segmentation. Accordingly, it isn’t clear how Cellos is an improvement on the status quo.

Optical clearing was used in multiple benchmark datasets and referred to throughout the manuscript. However, it wasn’t clear if this also an experimental parameter used in the experiments performed here? If so, a brief description of the clearing method used is warranted.

Multiple cell seeding ratios were used and Cellos was used to quantify the consistency in the ratio of co-cultured organoids. Does the ratio (measured by either mean intensity or sum intensity) give a consistent result? This was done in later sections but not here. If they are consistent, what is the true benefit over the simpler mean intensity approach?

Figure 3b. Labels should be added to the columns to show the seeding ratio, similar to what was done elsewhere in the figure.

Line 254 should mention the caveat that Cellos can perform this analysis presumably only when coupled with an engineered cell system.

Other methods have mentioned size and other morphological attributes can reduce performance of segmentation. Potential limitations and confounders of this method should be discussed.

Fig4 F. Statistical significance should be assessed. G and H can be merged into a single graphic rather than showing redundant controls.

Line 307, commonality should be supported with more than 1 reference.

Paragraph starting on line 320. Inclusion of a cellular viability dye to better assess the viability and morphologies of residual organoids in high concentration cisplatin treatment. This should at least be mentioned if not performed.

In addition to 80/20 train/test splits (line 340), split should be performed at the well or biological replicate level to show robustness to potential confounding experimental factors.

Line 454 should state the number of CPUs/threads used. Does the deployment of the method also rely on the utilization of a GPU?

Line 482, the ability to infer clonal heterogeneity is not something that is intrinsically enabled by Cellos alone and can be achieved using any method that accurately segment cellular regions. Consider revising wording.

Line 508. Avoid the use of superlatives. There have been multiple other papers that have presented a scalable single cell analysis approach for organoids/spheroids (for example <https://www.liebertpub.com/doi/full/10.1089/adt.2015.655>)

Line 618, what normalization was used (z, an external reference)?

Reviewers' comments and our responses below:

Reviewer #1 (Remarks to the Author):

The work by Mukashyaka, P., and team entitled “Cellos: High-throughput deconvolution of 3D organoid dynamics at cellular resolution for cancer pharmacology” is a showcase of a novel method to segment and analyze 3D models at cellular resolution with high throughput that is much required in the field of drug screening research, especially cancer drug screening.

The novel image analysis pipeline showed the accuracy and flexibility that can be used to deal with research questions at the organoids and the nuclei levels. The authors started by describing the TNBC model organoids with drug sensitivity heterogeneity (made from the PDX model TM00099) and derived and applied this cancer line to a few models to test the capability of the analysis pipeline.

Overall, this work demonstrated the advancement of the methods used for organoid-based drug screening, which is still inadequate and may be of interest to people who work in drug development and cancer modeling, etc.

We wish to thank the reviewer for these encouraging words.

However, there are some issues that may need further clarification.

Reviewer #1 Question 1. Has the pipeline been validated and tested in the other 3D cancer models, with different morphology such as the organoids with cyst-like, dispersed, compacted, or others? In order to generalize the application of this pipeline, data from 3 or more types of organoids should be included.

We agree that for the broader applicability of *Cellos*, it is important that its capabilities be tested on different types of organoids, which could present distinct segmentation challenges.

We therefore developed organoids from three cell lines: namely, HCC1806, MDA-MB231, and MCF10A. These cell lines were selected because they yield organoids of three different classically defined morphologies (*PMID: 18516279*): such as mass (compact), stellate (dispersed), and round and hollow when differentiated (cyst-like), respectively. See new Extended data fig. 2 for visual representation. To quantify the performance of *Cellos*, organoids were experimentally generated at three different cell seeding densities (seeding density1 (high), seeding density2 (medium), and seeding density3 (low)) for each of the three cell lines. The results of these analysis are described in the text below, which has been added into the Results section of the manuscript. The accompanying data is shown in Extended data fig. 2. The outcome confirms that *Cellos* is readily applicable to other cell lines with different morphologies.

“To test the broader applicability of *Cellos*, we determined if *Cellos* was effective on organoids from cell lines with different organoid morphologies. For this purpose, we analyzed a total of 426,810 cells from 11,416 organoids generated from TNBC cell lines HCC1806 and MDA-MB231 that have mass-like and stellate morphologies, respectively, as well as breast cell line MCF10A that has round and cyst-like organoids when differentiated (extended data fig.2 a-c). In this challenge, *Cellos* successfully segmented the organoids with distinct morphologies in a quantitative manner. In all cases, we observed a decrease in volume of segmented organoids with decreasing seeding densities of cells (extended data fig. 2d). Specifically, for HCC1806 the average organoid volumes ($\times 10^5 \mu\text{m}^3$) were 2.68 ± 0.01 , 2.65 ± 0.07 , and 2.17 ± 0.07 for high, medium, and low cell seeding densities respectively. For MCF10A the average organoid volumes ($\times 10^5 \mu\text{m}^3$) were 1.50 ± 0.20 , 1.39 ± 0.21 , and 0.96 ± 0.24 for high, medium, and low cell seeding densities respectively. For MDA-MB231 the average organoid volumes ($\times 10^5 \mu\text{m}^3$) were 2.99 ± 0.52 , 2.59 ± 0.48 , and 1.89 ± 0.28 for high, medium, and low cell seeding densities. Additionally, as expected, the number of cells per organoid segmented using *Cellos* decreases as the seeding density decreases, with p-value < 0.022 in all comparisons between high and medium seeding densities (p-value $< 2.551\text{e-}08$ in all comparisons, extended data fig.2e). Thus, we observe that *Cellos* effectively segments nuclei from organoids with distinct morphologies. Moreover, despite the variations in organoid morphologies and fluorescent dyes used for segmentation (see methods), no changes in *Cellos* pipeline parameters were required.”

The following text was added to the Methods section.

“3D culture and imaging for breast cell lines

TNBC cell lines MDA-MB231 (ATCC) were cultured in DMEM (Gibco) supplemented with 10% Fetal bovine serum (Gibco) and 1% Penicillin/Streptomycin (Sigma) and HCC1806 (ATCC) was cultured in DMEM (Gibco) supplemented with 15% Fetal bovine serum (Gibco) and 1% Penicillin/Streptomycin (Sigma). Breast cell line MCF10A (ATCC) was maintained in DMEM/F12 (Gibco) supplemented with 5% horse serum (Gibco), 1% Penicillin/Streptomycin (Sigma), 20 ng/ml EGF (Peprotech), 0.5 $\mu\text{g}/\text{ml}$ hydrocortisone (Sigma), 100 ng/ml cholera toxin (Sigma), and 10 $\mu\text{g}/\text{ml}$ insulin (Sigma)⁴⁴ and all cell lines were maintained in 5% CO_2 at 37°C. For 3D experiments, cells were seeded at varying densities on 35 μl of growth factor reduced Matrigel (Corning) per well of 96 well plates, allowed to grow for 4-5 days and imaged using the Opera Phenix system. For MDA-MB231, cells were seeded at increasing seeding densities of 2000 (low), 3000 (medium) or 4000 (high) cells per well and imaged after 4 days in 3D culture. HCC1806 and MCF10A cells each were seeded at three seeding densities (low, medium, and high) of 5000, 7000 or 10000 cells per well and 200, 500 or 700 cells per well respectively and

imaged after 5 days in 3D culture. MDA-MB231 and HCC1806 organoids were stained with Hoechst (Invitrogen) at final concentration of 5 $\mu\text{g/ml}$ and Calcein AM (Invitrogen) at a final concentration of 1 μM for 30 minutes at 37°C and imaged using above mentioned imaging conditions for a total of 165 individual zs per well. MCF10A organoids were stained with 5 $\mu\text{g/ml}$ Hoechst (Invitrogen) and imaged in a similar manner. For all cell lines and seeding conditions, images were collected from at least three and up to five replicate wells for each cell line and seeding density.”

Extended data fig.2: added into the manuscript is shown below.

Extended data fig.2: Cellos analysis on diverse organoid morphologies. 3D images of HCC1806 (a), MDA-MB231 (b) and MCF10A (c) organoids. Raw images are shown in the left panel, organoid segmentations via *Cellos* are shown in the middle panel, and a smaller field for better visualization of organoid segmentation is shown in the right panel. Individual segmented organoids are displayed in different randomly selected colors. **d.** *Cellos*-measured organoid volumes for three different cell seeding densities (seeding density1 (high), seeding density2 (medium), and seeding density3 (low)), for each of the three cell lines. Error bars represent standard deviation. **e.** *Cellos*-measured number of cells per organoid, for three decreasing cells seeding densities indicated as seeding density 1, 2, and 3 for each of the three cell lines. Error bars represent standard deviation.

Reviewer #1 Question 2. Interesting finding from the study is the ability of Cellos to reveal spatial relationships between cells in organoids. I strongly recommend that the finding be validated by other independent means.

Our understanding is that the reviewer is asking that the finding be validated through independent data analysis approaches. In our prior submission, we quantified clonal colocalization based on the proportion of cells of the same type around each cell c . This quantification was done as a function of window size K , i.e. among the K cells nearest in space to c , considered over all c .

As an additional approach, we have added a radial distribution analysis to calculate the co-localization of cells of the same type within a spatial radius r . This differs from the K -nearest-neighbors window approach in that the two methods are affected differently by irregularities in the density of cells within organoids, e.g. as would occur for organoids that are elongated or hollow. For the radial analysis, for each clone we calculated the *radial localization score*, which quantifies how often a cell's adjacent cells are of the same clonal type within a specified radius. The higher the localization score, the higher the co-localization of cells of the specified clone. Similar to the window-based localization score, the radial localization score is normalized for the clonal fraction in the organoid. Radial distribution analysis also demonstrated that cells of the same type tended to be co-localized. The data for this is shown in Extended data fig.8. e, f. Accompanying text has been added into the Results section of the manuscript and is shown below.

“We then used radial distribution as an alternate metric to assess the proximity of A50 cells to either B or another A50 cell (see methods). Radial distribution analysis quantifies the frequency of cells of each clonal type that are adjacent to cells of the same type within a specified radius. The higher the radial localization score, the higher the co-localization of cells of the specified clone. In such an analysis, we saw significantly higher clone localization in heterogeneously mixed organoids compared to homogeneously mixed organoids (Extended data fig. 8e, f). This was the case when comparing homogeneous B organoids to heterogeneous B-A50 organoids ($p=7.673e-5$) (Extended data fig.8e), as well as for comparison of homogeneous A50 organoids to heterogeneous A50-B organoids ($p=2.512e-5$) (Extended data fig. 8.f).”

The following text was added to the Methods section.

“Radial localization score calculation

To calculate the radial localization score of a fluorescently labeled clone, for example B-mCherry in an organoid: 1. The nuclear centroid was computed for each cell in the organoid. 2. All pairwise distances (Euclidean distance) between nuclei centers were computed. 3. The longest distance in the organoid was noted (r_{\max}). 4. For each B-

mCherry cell, the cells between $[0, r_i]$ were identified, where r_i was calculated for 6 equally spaced values from $r_{\max}/6$ to r_{\max} . 5. The proportion of B-mCherry cells was then calculated for each r_i and normalized by the proportion of B-mCherry cells in the organoid. The subsequent steps are the same as for the window-based co-localization analysis. Note for Extended data fig 8. e and f, results are shown at $r_i = 2(r_{\max}/6)$. Results at other values for r_i are qualitatively similar.”

Extended data fig. 8 e and f were added into the manuscript and are shown below.

Extended data fig.8. e. Distribution of radial localization score of B clones when mixed with A50 (blue line) or with alternately labeled B (orange line). **f.** Distribution of radial localization score localization score of A50 clones when mixed with B (blue line) or with alternately labeled A50 (orange line).

Reviewer #2 (Remarks to the Author)

The authors have developed a new computational method, called Cellos (Cell and Organoid Segmentation), for high-throughput segmentation and analysis of 3D organoids and their nuclei. The method uses classical algorithms to segment organoids in 3D and a Stardist-3D convolutional neural network to segment nuclei, allowing for analysis of cell densities, clonal population frequencies, and potential cell-cell interactions. Cellos was able to accurately distinguish ratios of distinct fluorescently labeled cell populations in organoids and recapitulate traditional luminescence-based drug response quantifications. The method was tested on triple negative breast cancer organoids containing clonal mixtures with complex cisplatin sensitivities, and was able to identify organoid and nuclear morphology feature

changes associated with treatment. Cellos provides powerful tools for high-throughput analysis for pharmacological testing and biological investigation of organoids based on 3D imaging.

We thank the reviewer for carefully reading our manuscript and providing helpful comments.

Comments.

Reviewer #2 Question 1. Several methods for segmenting organoids already exist, such as using automated segmentation methods like watershed segmentation, active contour models, and machine learning-based approaches. What is the advantage of Cellos compare to other existing methods? The authors should describe these points to prove the novelty of Cellos pipeline.

While there are several methods to segment organoids, there are few that, like *Cellos*, can take 3D input data and then segment the organoids in 3D. Most prior algorithms fall into four categories with clear deficiencies compared to *Cellos*: 1. Images are taken in 2D and the downstream analysis including segmentation is done in 2D (eg: CALYPSO [1], OrgaQuant [3]); 2. Images are taken in 3D and segmentation is done on 2D maximum projection images (eg: OrganoSeg [2], Spiller, Erin R et al. [5]), 3. Images are taken in 3D and analysis is done on 2D average stack projections (e.g.: MorgAna [6]). In these first 3 categories, all methods do not use volumetric information in the segmentation step. In a 4th category, images are taken in 3D, but the analysis is done at the voxel level without segmentation (eg: Phindr3D [9]).

To the best of our knowledge, there are three methods that perform 3D volumetric organoid segmentation (Beghin et al. [11], Boutin et al. [12], and Zhang, Linyi et al. [10]). We cited two of these studies in the prior submission and noted their inability to handle multiple organoids per well “Boutin et al. developed a 3D spheroid and nuclei segmentation approach for optically cleared images of one single spheroid in a well. More recently, Beghin et al. developed a segmentation technique for the Jewell system again with one organoid per well”. We have further identified Zhang, Linyi et al. [10], which describes a method optimized for segmenting organoids from optical coherence tomography images, but this method lacks cellular resolution. The distinctiveness of *Cellos* is that it segments large numbers of organoids at cellular resolution within each well and in true 3D. Combined, these aspects enable accurate high-throughput quantification with little user-specified parameter tuning.

We have added the following text to the introduction section.

“Zhang, Linyi et al¹⁸ also describes a method optimized for segmenting organoids from optical coherence tomography images but this method lacks cellular resolution.”

We have added the following text to the discussion section.

“The distinctiveness of *Cellos* compared to other organoid segmentation methods is that it segments large numbers of organoids at cellular resolution within each well in flexible culture conditions and in true 3D. Combined, these aspects enable accurate high-throughput quantification with little user-specified parameter tuning compared to other extant methods (supplemental file, table 1).”

We have added the following table to supplemental file.

Pipeline	Multiple organoids per well	3D/2D segmentation
CALYPSO [1]	Yes	2D
OrganoSeg [2]	Yes	2D
OrgaQuant [3]	Yes	2D
Larsen, Brian M et al. [4]	Yes	2D
Spiller, Erin R et al. [5]	Yes	2D
MOrgAna [6]	No	2D
Hof, L., Moreth, T., Koch, M. et al. [7]	Yes	2D
OrganoID [8]	Yes	2D
Phindr3D [9]	Yes	2D
Zhang, Linyi et al. [10]	Yes	3D
Beghin et al. [11]	No	3D
Boutin et al. [12]	No	3D
Cellos	Yes	3D

Supplemental file. Table 1. Comparison of *Cellos* and other organoid segmentation techniques.

References:

[1] Bulin, Anne-Laure et al. “Comprehensive high-throughput image analysis for therapeutic efficacy of architecturally complex heterotypic organoids.” Scientific reports vol. 7,1 16645. 30 Nov. 2017.

[2] Borten, M.A., Bajikar, S.S., Sasaki, N. et al. Automated brightfield morphometry of 3D organoid populations by OrganoSeg. Sci Rep 8, 5319 (2018).

[3] Kassis, T., Hernandez-Gordillo, V., Langer, R. et al. OrgaQuant: Human Intestinal Organoid Localization and Quantification Using Deep Convolutional Neural Networks. Sci Rep 9, 12479 (2019).

[4] Larsen, Brian M et al. “A pan-cancer organoid platform for precision medicine.” Cell reports vol. 36,4 (2021): 109429.

[5] Spiller, Erin R et al. “Imaging-Based Machine Learning Analysis of Patient-Derived Tumor Organoid Drug Response.” Frontiers in oncology vol. 11 771173. 21 Dec. 2021

[6] Gritti, Nicola et al. “MOrgAna: accessible quantitative analysis of organoids with machine learning.” Development (Cambridge, England) vol. 148,18 (2021): dev199611.

- [7] Hof, L., Moreth, T., Koch, M. et al. Long-term live imaging and multiscale analysis identify heterogeneity and core principles of epithelial organoid morphogenesis. *BMC Biol* 19, 37 (2021).
- [8] Matthews, Jonathan M et al. "OrganoID: A versatile deep learning platform for tracking and analysis of single-organoid dynamics." *PLoS computational biology* vol. 18,11 e1010584. (2022).
- [9] Mergenthaler, Philipp, et al. "Rapid 3D phenotypic analysis of neurons and organoids using data-driven cell segmentation-free machine learning." *PLOS Computational Biology* 17.2 (2021): e1008630.
- [10] Zhang, Linyi et al. "Quantifying the drug response of patient-derived organoid clusters by aggregated morphological indicators with multi-parameters based on optical coherence tomography." *Biomedical optics express* vol. 14,4 1703-1717. (2023).
- [11] Beghin,A.,Grenci,G.,sahni,G. et al. Automated high-speed 3D imaging of organoids cultures with multi-scale phenotypic quantification.*Nat Methods* 19, 881-892(2022).
- [12] Boutin, M.E., Voss, T.C., Titus, S.A. et al. A high-throughput imaging and nuclear segmentation analysis protocol for cleared 3D culture models. *Sci Rep* 8, 11135 (2018).

Reviewer #2 Question 2. Figure 1A: the relative viability of untreated B- and A50-clones is different (~1.1 in B-clone and ~0.9 in A50-clone) at the starting point. What is the value normalized to? Is cell proliferation different between these clones? A slow-growing phenotype can affect the drug sensitivity. The author should characterize the proliferation ability of A50- and B-clone cells not only for the pure organoids but also homogeneously or heterogeneously mixed ones.

For the IC₅₀ curves in Figure 1A, for each sample we normalized the luminescence signals for each drug concentration to control. This allowed us to determine relative viability of the cells after exposure to each of the drug concentrations. Doing this sets relative viability of the control to "1" for each series. Since the x-axis in the IC₅₀ plot is a log scale, we cannot plot the control condition where drug concentration is "0". Hence the starting point of the curve is the relative viability at the lowest dose of drug, in our case 0.5 μM Cisplatin. The values of ~1.1 in B clone and ~0.9 in A50 clone correspond to their relative viabilities at this 0.5 μM Cisplatin concentration respectively.

The reviewer brings up a good point that growth rates of different cells can affect cisplatin response. Unfortunately, due to image acquisition times in our setup it is not possible to obtain frequent enough time points to directly observe proliferation. This is because it takes several hours for a single scan of a plate, and the cells cannot survive multiple imaging scans in the high content imaging device (they are grown externally and are transferred to the imager prior to measurement). This currently constrains our ability to obtain sequential images in time. Hence, unfortunately we cannot comment on growth rate effects in the manner suggested by the reviewer. Since this is a

limitation of the hardware of the image capture system, our algorithmic software cannot resolve this problem. However, if such sequential images can be made available, *Cellos* can be deployed to assess growth rates as a measure of drug response as suggested since the metrics required for this analysis would be organoid volumes and cell numbers at different timepoints, both of which can be obtained easily via *Cellos*.

We have changed the following text in the discussion “*Cellos* enables interpretation of these parallel measurements by individually quantifying cells and organoids to elucidate chemical biology and pharmacology effects.” to “*Cellos facilitates interpretation of these parallel measurements in organoids by individually quantifying and analyzing organoids and their cells in 3D to elucidate growth and pharmacology effects.*”

Reviewer #2 Question 3. Figure 1B: scale bars are missing.

We have incorporated scale bars to the images. Additionally, we have added 3D representations of the images to highlight that the pipeline analyzes images in three dimensions. The updated Figure 1 is shown below.

Figure 1. Outline of *Cellos* pipeline and cellular system. **a.** Cisplatin IC₅₀ curves for two clones A50 (blue line) and B (gray line) from 3D homogeneously mixed organoids using cell-destructive luminescence readout. **b.** *Cellos*: Two-stage pipeline for 3D organoids and nuclei segmentation on 3D images. **c.** Top panel exhibits steps for 3D organoid segmentation. The inputs are 3D z-stack images, and the outputs are the segmented and labeled organoids, bottom panel shows an example of 3D organoids before and after segmentation. **d.** Steps for nuclei segmentation. A Stardist-3D with Resnet backbone model²² is trained using the training dataset. The trained model is then applied to experimental data with individual segmented organoids as input, and segmented and labelled nuclei as outputs. The 3D figures were generated using napari²⁴ python library, which has been integrated into *Cellos* pipeline.

Reviewer #2 Question 4. Line 90: What ‘primary cell lines’ means? ‘primary’ and ‘cell line’ are contradictory. Do the authors use multiple cell lines or is it ‘a cell line’ not ‘cell lines’?

Thank you for pointing this out. The terminologies for the different kinds of primary cultures are indeed getting more confusing with new culture conditions and technologies. For example, organoid cultures can grow cancer cells almost indefinitely (like a cell line) even though when placed on plastic (as with standard cell lines) they do not survive. For this reason, others have used the term, primary cell lines (PMID: 34395440). However, for the purpose of this paper, we will use the terminology of “primary cell cultures” to avoid any ambiguities.

We have changed the following text “As a case study for 3D image analysis, we generated organoids from primary cell lines derived from a TNBC Patient Delivered Xenograft (PDX) model TM00099¹⁸” to “As a case study for 3D image analysis, we generated organoids from primary cell cultures derived from a TNBC Patient Delivered Xenograft (PDX) model TM00099¹⁸”. And we have also changed “After *in vitro* expansion of the primary cell lines, single human cells labeled with Anti-Human HLA-ABC APC (eBiosciences) were isolated using flow cytometry and further expanded to establish clonal lines” to “After *in vitro* expansion of the primary cell cultures, single human cells labeled with Anti-Human HLA-ABC APC (eBiosciences) were isolated using flow cytometry and further expanded to establish clonal lines.”

Reviewer #2 Question 5. The authors show the high accuracy of *Cellos* nuclei segmentation by the use of only one external dataset (breast carcinoma from Boutin et al.; 184). The authors should investigate if the results can be reproduced in another available dataset.

We have now additionally also applied *Cellos* to segment nuclei from a 3D image of a mouse embryo (<https://doi.org/10.1093/bioinformatics/btaa029>) with manually annotated nuclei. *Cellos* was able to segment 52 nuclei out of the 56 annotated nuclei in the mouse embryo. This image

was taken at 40x magnification, so we had to reduce the magnification to 20x to be compatible with *Cellos*.

We also ran *Cellos* on a synthetic dataset of leukemia cell line HL60 from (<https://bbbc.broadinstitute.org/BBBC024>). The dataset contains four synthetic image subsets each containing 30 images (with each image containing 20 HL60 cell line nuclei). The four subsets are based on different probabilities (0%, 25%, 50%, and 75%) of the nuclei clustering, representing how close or far apart the nuclei are in the image, with higher probabilities indicating the nuclei are densely packed. In addition, each subset has both low and high signal to noise ratio (SNR) images. We were able to accurately segment the nuclei in all the images regardless of SNR and nuclei cluster probabilities. In all cases precision, recall and accuracy were greater than 0.95.

The following text has been added to the Results section of the manuscript.

“Additionally, we applied *Cellos* on a leukemia cell line HL60 synthetic dataset from²⁵ consisting of four synthetic image subsets each containing 30 images of different nuclei densities and signal to noise ratios. *Cellos* was able to segment the nuclei with precision, accuracy, and recall of > 0.95 in all subsets. Finally, we were also able to segment 52 out of 56 annotated nuclei from an image of a mouse embryo²⁶. Thus, *Cellos* was able to process external image datasets with quantitative precision.”

Reviewer #2 Question 6. Can Cellos discriminate between dead cells, quiescent or dormant cells, senescence cells, or proliferating cells by detecting the nuclear fluorescence labeling after drug exposure?

We observe differences in the patterns of nuclear fluorescence for dead cells. To prove this, we first defined a gold standard set of dead cells based on the overlapping signal between segmented nuclei and dead cell dye DRAQ7. We then analyzed the nuclei morphology of HCC1806 breast cancer cells treated with docetaxel, where we could distinguish live vs. dead cells using DRAQ7 overlapping signal. We saw significantly lower mean fluorescence intensity of Hoechst (p-value = 3.0e-7) and bigger nuclei (p-value=0.002) in dead cells (Rebuttal Figure 1).

Rebuttal Figure 1. Analysis of nuclei volume (left panel) and mean intensity of Hoechst (right panel) in live cells and dead cells in HCC1806 organoids treated with docetaxel.

We agree other cell states are interesting, but it is non-trivial to obtain accurate labels for them as this would require additional experiments, optimization for other markers, and analysis beyond the scope of this work. For example, it would be interesting to assess proliferating cells via investigations of staining with either Ki67 antibody or by BrDU, but the analysis of whether there are signal thresholds associated with morphology is affected by uncertainties in the underlying cell biology. Concerning dormancy, the B clone exhibits a dormant/persistent like phenotype at high platinum doses, and we do see a small perceivable difference in nuclear morphology. To ascertain whether this morphology is linked to the dormant state or is simply a morphological difference between subclones would require significantly more open-ended investigation.

Reviewer #2 Question 7. The authors should apply the Cellos method to quantify the treatment response to other patient-derived tumor organoids. Moreover, the degree of cell damage or death induced by the drugs varies depending on the type of drug. Therefore, other drug treatments should be evaluated in addition to cisplatin.

To address this question, we analyzed organoids from the TNBC cell line HCC1806 treated with Docetaxel, Doxorubicin and Cisplatin, both at the organoid and cellular level. We specifically chose Docetaxel and Doxorubicin since both these drugs are chemotherapeutic agents commonly used for the treatment of TNBC tumors, and their mechanisms of action differ from Cisplatin. To quantify the accuracy of *Cellos* for each drug, we treated the organoids with two different doses (dose1 < dose2) or control, in triplicate conditions. We also intentionally chose high doses of the drugs, as they would contain more debris and would thus be more challenging to segment (extended data fig. 5a).

Organoid level analysis demonstrated a decrease in organoid volume as the drug concentration increased (extended data fig. 5b), with the organoid volume from the control being significantly

larger than the treated organoids for all the drugs (p-value = 4.121e-100). This shows that indeed *Cellos* is able to effectively segment organoids treated with different drugs. For the cellular level analysis, we compared the number of cells detected by *Cellos* to the respective luminescence assay signals and observed that they are highly correlated (correlation=0.962 with 95% confidence interval = 0.916 - 0.982) (extended data fig. 5c).

We added the following text to the manuscript in the Results section to include this data.

“To determine if organoid and nuclear segmentation post drug treatment would be efficient in a different cell line model and using different therapeutic agents, we used *Cellos* to analyze organoids generated from TNBC cell line HCC1806 treated with cisplatin or two additional chemotherapeutic agents commonly used for TNBC treatment, namely Docetaxel and Doxorubicin (extended data fig. 5a). *Cellos* was used to segment HCC1806 organoids and yielded a monotonic decrease in organoid volume as the drug concentration increases for all three drugs (extended data fig. 5b). Additionally, the number of viable nuclei counted using *Cellos* was highly correlated with luminescence assay signals acquired post imaging (correlation=0.961, 95% confidence interval = 0.916 - 0.982) (extended data fig. 5c). *Cellos* was effective in organoid and nuclear segmentation despite the organoids being treated with relatively high drug doses, which creates abundant cell debris that could in principle have interfered with segmentation (extended data fig. 5a). Note that segmented nuclei that were also positive for dead cell stain DRAQ7 were identified as dead cells and removed from the analysis. These results show the ability of *Cellos* to segment organoids and nuclei generated from an established cell line treated with three drugs with distinct mechanisms of action.”

We added the following text to the methods section.

“HCC1806 3D organoids drug screen

For drug experiments on the HCC1806 cell line, 7000 cells were plated on 35 µl of growth factor reduced Matrigel (Corning) per well of 96 well plates on day -3 and organoids were allowed to form for three days. On day 0, media was replaced, and the cells were treated with two drug concentrations (Dose1 and Dose2) each for three drugs namely, Cisplatin, Doxorubicin and Docetaxal in triplicate conditions. Organoids were treated with 15 µM and 30 µM Cisplatin (Selleck Chemicals), 1 µM and 2 µM Doxorubicin (Selleck Chemicals) and 6.5 nM and 13 nM Docetaxel (Selleck Chemicals) respectively. Organoids were maintained with drugs for 72 hours, stained with 5µg/ml Hoechst (Invitrogen), 1µM Calcein AM (Invitrogen) and 1µM DRAQ7 (Abcam), incubated for 30 minutes at 37°C and imaged using above mentioned imaging conditions for a total of 200 individual zs per

well. CellTiter-Glo® 3D Cell Viability Assay (Promega) was then used for luminescence-based assay readouts using manufacturer's protocols.

Detecting and removing dead nuclei from analysis

Cellos includes a customized step to detect dead nuclei and remove them from the subsequent analysis. To do this, for each individual cropped organoid, we first normalize its dead cell dye (DRAQ7) channel intensity so that the intensity is between 0-1. This is performed because intensity values may vary across distinct organoids. We then proceed to evaluate each segmented nucleus within an organoid. We calculate the average intensity of the DRAQ7 channel in the segmented nucleus bounding box region. The resulting calculated number is then used to categorize each nucleus as live or dead. To make this classification we rely on a predetermined threshold of nucleus with intensity > 0.1 as dead. The threshold was chosen by evaluating many histograms illustrating the average DRAQ7 intensity of all the nuclei in several different wells.”

Extended data fig. 5 added in the manuscript is shown below.

Extended data fig.5: *Cellos* analysis of organoids treated with chemotherapy drugs. **a.** 3D images showing HCC1806 organoids treated with different drugs from left to right (control, cisplatin-dose2, Docetaxel-dose2, Doxorubicin-dose2). **b.** Barplot depicting organoid volume changes at different doses of the three drugs. The error bars are from three replicates for each condition. A total of 14,250 organoids were analyzed. **c.** Correlation plot of total cells counted by *Cellos* versus

the luminescence signal for individual wells. A total of 271,011 segmented cells were used for this analysis.

Reviewer #2 Question 8. In the cisplatin treated case, damaged nuclei would be much harder to be segmented as the deformity and intensity of the fluorescence substantially varies. Is there any threshold to take the weak intensity of fluorescence into account as nuclei? The images of cisplatin-treated A-50 cell should be shown in Figure 5 not just of the B-mCherry since A-50 would be much more damaged by cisplatin in higher doses.

We agree that the fluorescence intensity of nuclei indeed decreases after treatment as shown in extended data fig. 7c. We know there is variation in fluorescence intensity of annotated nuclei in our training data, as we ensured that they were included in the manual annotation of 24 drug-treated organoids. In addition, we do not use an absolute threshold to account for nuclei with low intensity, because the intensity of each individual channel in each organoid is normalized before the nuclei segmentation step. This normalization is percentile based and is part of Startdist⁴⁵. Thus, the segmentation should not be affected by absolute fluorescence intensity variation. We believe our approach is a better way to normalize the dataset than to apply thresholds since there often are variations in fluorescence intensities between different experiments. The fact that nuclear intensity does decrease post-drug simply provides another biological parameter to interrogate if one chooses to. To illustrate these points, below we have provided example images showing that *Cellos* can segment nuclei with low and high fluorescence intensity (supplemental file Figure 1.d).

To address this point, the following text has been added to the method section.

“It is important to note that *Cellos* can segment nuclei regardless of their fluorescence intensity (supplemental file Figure 1.d).”

Images to support this are shown below and have been added in supplemental file Figure 1.d.

supplemental file Figure 1.d: Nuclei segmentation of EGFP and mCherry labeled nuclei. 3D grid visual representation of segmentation of EGFP and mCherry labeled nuclei with various fluorescence intensities. Separate example organoids are depicted in the two panels.

Regarding the reviewer's questions about A50, it is true that A50 would be more damaged at higher concentration of cisplatin. As per reviewer suggestion, we have moved images showing the change of A50 nuclei from the extended data figure to the main text figure 5h.

Reviewer #2 Question 9. *The authors should show nuclei segmentation images for cisplatin treated case like in Figure 2. More visible pictures should be provided as supplementary information. The pictures in Figure 2a are too small to judge the performance of Cellos.*

We have increased the size of the images in Fig 2a and added a supplemental file showing performance of *Cellos* on segmenting organoids and nuclei in cisplatin treated cases (supplemental file Figure 1-4)

The following text has been added to the Results section of the manuscript.

“For more organoid segmented images see supplemental file Figure 1-4(a-c).” And
“For additional visual inspection of nuclei segmentation see supplemental file Figure 1-4(d).”

Reviewer #2 Question 10. Line 201: *What day0 means? Immediately after making mixture of cell suspensions? Several hours after waiting for spheroid formation? It takes time for the suspended cells to aggregate and further form spheroids. The authors should show the method*

of making chimeric spheroids more in detail. Time course study of spheroid formation from cell suspension should be performed. Please show the day0 images in Figure 3.

As described in our methods section, we seed single cells on day “-3”, allow for cells to aggregate and/or grow to form organoids for three days, then add drug at day 0. So, day “0” in the text is with respect to the day we can see well-formed organoids and the addition of drug.

For clarification, we have changed “These organoids were imaged at day0, allowed to grow, and imaged again 4 days later (day4) (see methods dataset-2 for details).” to “**These organoids were seeded on day “-3” imaged at day0 (after organoids were formed, extended data fig. 3a), allowed to grow, and imaged again 4 days later (day4) (see methods dataset-2 for details).**”

We have also added in these day0 images with well-formed organoids for clarity in extended data figure 3.a, which is also shown below.

Extended data fig.3.a Representative z-axis maximum projection images of homogeneously mixed organoids generated with seeding percentages of 20%, 40%, 60% and 80% A50-EGFP, respectively, with the remaining cells being A50-mCherry. Images are from day0, and scale bar represents 100 μm .

Organoid formation *in vitro* is a dynamic biological process that has been well characterized over the years. For example, Wang et al., 2012 show time lapse images of MCF10A breast organoid formation from a single cell over a 10-day time period (PMID: 23248267; Figure 1A and Movie S1). *In vitro* self-organization of organoids consisting of multiple cell types has been studied in great detail in organoids, for example in mammary epithelial cells (PMID: 25633040) and skin epithelial cells (PMID: 28798065), respectively.

Via brightfield imaging we also observe that our cells aggregate and start to form organoids by day 1 and grow over time (Rebuttal Figure 2). Unfortunately, we did not have the ability to generate time lapse movies using our current hardware that does not accommodate the live monitoring of the cells within the imaging device over time.

Rebuttal Figure 2. Representative brightfield images of single planes showing organoid formation from single cells starting 2 hours after cell seeding (on day -3), day -2, day -1, day 0 and day 4.

Reviewer #2 Question 11. *Figure 5: g and f are mislabeled.*

Thank you for pointing this out. We have made this revision accordingly.

Reviewer #2 Question 12. *Spatial relationship in Figure 6 is intriguing and provides a new biological insight, although dividing the possible cause into 2 mechanisms, cell division and ecological affinity, is too simplified. It is possible that the cells attaching to the similar cells are more advantageous for cell division. Do the heterogeneously mixed cells move in the spheroids to attach to the similar cells each other? Time lapse study would provide useful information. Can Cellos be applied to such analysis? The nuclei segmentation would be useful for the precise tracking of the nuclei.*

All these questions raised are indeed very interesting, however, to address these biological questions require a different experimental design with selectively engineered cells and different image capture platform that allows for time lapse photography. We have provided the analytical framework to address these questions, so in presence of such data *Cellos* can be updated to track nuclei in 3D based on their centroids, bounding box, and morphologies. The proper experiments to address primary biological questions as posed is beyond the intent or the scope of this manuscript.

Reviewer #2 Question 13. The potential limitations of this approach, and how they might affect the applicability should be discussed. Please discuss the following points:

a) Are there any potential errors or inaccuracies in nuclei segmentation due to the high density of cells in organoids?

Thank you for an important question. To answer this, we used the Broad Bioimage Benchmark Collection (<https://bbbc.broadinstitute.org/BBBC024>) dataset which has annotated biological image sets for testing and validating segmentation techniques. Please refer to the response to your *Question 5* for further details on the dataset and results. We did not observe any significant variations in nuclei segmentation accuracy across images with different densities of nuclei. Therefore, we conclude that the performance of *Cellos* is not affected by density of cells in this dataset.

b) What are the potential limitations of using a training dataset of only 36 organoids to develop a CNN for nuclei segmentation?

Even though we only used 36 organoids, in total they consisted of 3,862 individual nuclei, which were augmented using elastic deformation, noise/intensity shift, and flip / 90-degree rotation. Thus, including raw images, 15,448 nuclei were used. Moreover, we did not see significant changes in validation metrics when we used 36 images versus 30 images in the CNN model. For the CNN model with 30 images, precision= 88.15 ± 4.7 , recall = 87.47 ± 3.76 and accuracy = 78.31 ± 6.09 , while for 36 images, precision= 88.85 ± 3.24 , recall= 87.73 ± 3.55 and accuracy= 79.08 ± 4.49 .

To address this point, the following text has been added to the method section.

“Note that we did not see significant changes in validation metrics when in total 30 organoids were used (precision= 88.15 ± 4.7 , recall = 87.47 ± 3.76 and accuracy = 78.31 ± 6.09).”

c) Are there any potential biases in using specific fluorescent labels (EGFP, mCherry or Hoechst) for nuclei segmentation in this study?

Among 36 organoids we manually annotated, 10, 12 and 14 organoids were fluorescently labeled with Hoechst, EGFP and mCherry respectively. These organoids were chosen to have a good representation of all the fluorescent labels in the training dataset. We did not see any biases towards a particular fluorescent label. Moreover, from all the experimental datasets (for example: see extended data table 1) we analyzed where we used EGFP or mCherry, we did not detect positive or negative biases towards either fluorescence label.

d) While the pipeline is described as high-throughput, what are the potential limitations or bottlenecks? How long actually does it take to analyze large datasets of organoids?

Thank you for the question, we apologize for not clarifying this. In the discussion we said that “Altogether, when pre-processing and morphological feature characterization are included, it took on average ~1.9 hours with CPU efficiency of 91.2% and ~100 GB of computational memory to segment 550 organoids, and ~1.4 hours with CPU efficiency of 82.31% and 6.86 GB of computational memory to segment 4837 nuclei.”

To clarify the potential limitation, we have added this point to the Discussion “**This was possible because the pipeline is optimized to work on high performance computing systems.**”

Reviewer #3 (Remarks to the Author)

Synopsis. The authors fine-tuned a single cell segmentation method for organoid/spheroid cultures using a deep learning-based method (Stardist/RESNET). The method/model, referred to as Cellos, was trained on a curated dataset and tested on a larger library of spheroid images generated elsewhere. Conventional feature extraction was performed and used for downstream analysis. Use applications included quantification of clonal architecture and response to drugs, as well as identification of features that were used to classify genotoxic drug mechanism of action. The methods and experimental approaches implemented here were done with rigor and followed industry standards. Example data and analytical scripts are made available in a well-documented/structured GitHub location that supplements the text. My only concerns are that the author should place more emphasis on what truly differentiates their approach from previous generations of analysis.

We thank the reviewer for carefully reviewing our manuscript and additional documentation.

Itemized list of concerns:

Reviewer #3 Question 1. On line 150, manual quantification was performed as a benchmark. Was only one “expert” used or did multiple experts quantify the data to evaluate operator error?

Thank you for your question. To mitigate this potential problem, we had two individuals manually quantify the data and the final annotations were decided by consensus.

Reviewer #3 Question 2. LabKit, a more conventional ML-segmentation approach, was used to be “ground truth” to benchmark Cellos. The data presented shows a generally high IoU between

the methods, indicative of comparable segmentation. Accordingly, it isn't clear how Cellos is an improvement on the status quo.

We apologize for the confusion. LabKit, which is a plugin in ImageJ, was used solely to facilitate manually annotating the data, then the annotated data was used to train a model that was subsequently used to segment nuclei in the experimental images.

Reviewer #3 Question 3. Optical clearing was used in multiple benchmark datasets and referred to throughout the manuscript. However, it wasn't clear if this is also an experimental parameter used in the experiments performed here? If so, a brief description of the clearing method used is warranted.

The relevant section mentions that “We used data from the Boutin, et al.¹⁵, who manually annotated cells in three optically cleared spheroids grown and assayed in a manner distinct from our platform”. This analysis was to show that despite their specialized culture system, *Cellos* was able to accurately segment their data. Optical clearing was not used to generate our training data or in any of our experiments. The results on the Boutin et al data demonstrate the robustness of *Cellos* to data generated in different experimental systems.

To avoid any confusion, we changed the following text “We used data from the Boutin, et al.¹⁵, who manually annotated cells in three optically cleared spheroids grown and assayed in a manner distinct from our platform” to “**We used the Boutin, et al.¹⁵, who manually annotated cells in three optically cleared spheroids grown and assayed in a manner distinct from our platform which does not use optical clearing.**”

Reviewer #3 Question 4. Multiple cell seeding ratios were used and Cellos was used to quantify the consistency in the ratio of co-cultured organoids. Does the ratio (measured by either mean intensity or sum intensity) give a consistent result? This was done in later sections but not here. If they are consistent, what is the true benefit over the simpler mean intensity approach?

The high reproducibility of cell ratios in co-cultured organoids was shown in Figure 3, using cell counts per well in Figure 3b and cell counts per organoid in Figure 3c. The reviewer is correct that mean intensity per organoid is similar to the *Cellos* cell counting based ratios. For example, we see a decrease in mean EGFP intensity in organoids in wells with decreasing pre-determined percentages of EGFP+ cells, as well as a contrasting increase in mean mCherry intensity per organoid as shown below (Rebuttal Figure 3).

The fundamental advantage of *Cellos* is that it yields segmentations and locations of each organoid and cell nucleus. It also provides measurements of organoid shape, nuclear shape, and cell spatial

distributions. All these metrics are potentially valuable to investigating biological responses to drugs, stimulants, etc. Such behaviors are not captured in simple metrics such as mean intensity.

Rebuttal Figure 3. Box plots showing mean intensity of EGFP (left panel) and mCherry (right panel) in organoids in wells with pre-determined ratios of EGFP:mCherry cells.

Reviewer #3 Question 5. Figure 3b. Labels should be added to the columns to show the seeding ratio, similar to what was done elsewhere in the figure.

As the reviewer has suggested, we have added the seeding ratios to the figure 3b.

Reviewer #3 Question 6. Line 254 should mention the caveat that *Cellos* can perform this analysis presumably only when coupled with an engineered cell system.

Fluorescently labelled nuclei are required for nuclear segmentation by *Cellos*. For most of our experiments, we labelled our cells with nuclear-expressing EGFP or mCherry because we wanted to distinguish the differential behavior of two related clones. However, we also trained our model on commonly used fluorescent nuclear dyes such as Hoechst, which can be assayed without cell engineering. Moreover, in our newly generated data (Extended data fig 2 and extended data fig 5) we show that Hoechst can be used by *Cellos* to segment and quantify nuclei.

We have made edits to line 254 from: “These observations were consistent with the standard luminescence assays (Fig.1a), indicating that *Cellos* can recapitulate those findings by counting individual cells of specific cell types, but in a non-destructive and higher (cell) resolution manner.” to “These observations were consistent with the standard luminescence assays (Fig.1a), indicating that *Cellos* can recapitulate those findings by counting individual fluorescently labelled nuclei of specific cell types, but in a non-destructive and higher (cell) resolution manner.”

We have also added the following text in the discussion “While we genetically engineered the expression of fluorescent tagging in many of our experiments to distinguish the

behavior of different cancer clones, nuclear dyes such as Hoechst can also be used for segmenting nuclei thus broadening *Cellos*' applicability.”

Reviewer #3 Question 7. Other methods have mentioned size and other morphological attributes can reduce performance of segmentation. Potential limitations and confounders of this method should be discussed.

Thank you for bringing this up. For the organoid segmentation step, the main limitation is misidentification of debris as organoids. This is seen predominantly at high treatment concentrations, as debris arise from dead cells. This was described in the prior submission, relevant to Extended data fig. 6a-d. Additionally, at high treatment concentrations, there may be greater numbers of single cells living independently. These would not be identified as organoids in our standard pipeline because of the minim cutoff we applied on organoid size to exclude debris from being segmented. This can be corrected by adjusting the lower limit on the size of organoids during organoid segmentation based on the requirements of the study. We have added image examples to visualize these two cases (supplemental file Figure 5).

To address this, the associated text was changed from “This suggests that the structure of the large organoids appears to be maintained despite a reduction in cellular content (as measured by lower cell density, solidity, and higher Euler number)” to “**This suggests that the structure of the large organoids appears to be maintained despite a reduction in cellular content (as measured by lower cell density, solidity, and higher Euler number), and an increase in debris arising from dead cells (supplemental file Figure 5a)**”

The follow text was also added to the manuscript.

“In this case at high doses of drug, the recall value for organoid segmentation is lower than precision because of the cutoff we used on minimum organoid size. The cutoff which was chosen to reduce segmented debris inevitably also excluded some of the single cells from being segmented as organoids as shown in supplemental file Figure 5b.”

Supplemental file Figure 5. Image showing example of organoid segmentation limitation after exposure to high concentration of drug (Cisplatin =128 μ M). **a.** circled region shows debris segmented as organoids, and **b.** circled region shows single cells not segmented as organoids.

For the nuclei segmentation step, the main limitation of the method is the shape of the nucleus to be segmented. The nucleus has to be star-convex, meaning that any ray extending from the center reaches the boundary once⁴⁵, otherwise they cannot be segmented properly.

The following text was also added to the method section.

“It is important to note that non-star-convex-shaped nuclei cannot be segmented properly by the model.”

Reviewer #3 Question 8. Fig4 F. Statistical significance should be assessed. G and H can be merged into a single graphic rather than showing redundant controls.

Thank you for your comments.

The following text has been changed from “The median A50-EGFP percentage across organoids increased from 52 % to 72 % from 0 to 2 μ M and decreased to 23 % at 64 μ M, with variability amongst individual organoids (Fig.4f and g)” to “The median percentage of A50 EGFP cells per organoid in control untreated wells, was 57% which increased to 72% at 2 μ M ($p=2.774e-13$) and decreased to 23% at the high cisplatin concentrations of 64 μ M ($p= 3.142e-56$, Figure 4f).”

Per reviewer’s recommendation, we have merged Fig. 4G and 4H into one figure Fig. 4G shown below.

Figure. 4g Percentage of A50-EGFP cells per organoid versus total cells. Each dot represents an organoid. White dots represent untreated organoids, and blue and orange dots represent organoids treated with 2 μ M and 64 μ M cisplatin respectively. Median percentage of A50-EGFP cells for each condition are marked by the triangles below.

Reviewer #3 Question 9. Line 307, commonality should be supported with more than 1 reference.

Two additional references utilizing organoid morphology analysis have been added in the manuscript namely^{28,29}.

28. Matthews, Jonathan M et al. “OrganoID: A versatile deep learning platform for tracking and analysis of single-organoid dynamics.” PLoS computational biology vol. 18,11 e1010584. 9 Nov. 2022,
29. Spiller, Erin R., et al. "Imaging-based machine learning analysis of patient-derived tumor organoid drug response." Frontiers in oncology 11 (2021): 771173.

Reviewer #3 Question 10. Paragraph starting on line 320. Inclusion of a cellular viability dye to better assess the viability and morphologies of residual organoids in high concentration cisplatin treatment. This should at least be mentioned if not performed.

We agree with the reviewer that using a viability dye would help to better assess the morphologies of organoids in presence of high concentration of cisplatin treatment. However, this was not performed in our experiments to avoid channel leakage due to the overlapping emission spectrum range of the dyes used. Regarding this point, we have added the following text to the Discussion.

“Additionally, use of viability dyes can be beneficial in better assessing the morphologies of organoids and cells when exposed to high concentration of drug.”

Reviewer #3 Question 11. In addition to 80/20 train/test splits (line 340), split should be performed at the well or biological replicate level to show robustness to potential confounding experimental factors.

We agree with the reviewer that the images used to train/test the model should come from diverse wells and biological replicates. This is indeed what we did as stated in the methods section “In total 36 organoids were labeled. 10, 12 and 14 organoids had cells which were fluorescently labeled with Hoechst, EGFP and mCherry respectively. Hoechst labeling was performed to diversify the training set. Among these 36 organoids, 24 had been treated with cisplatin.”

We changed the first sentence to reflect this point from “In total 36 organoids were labeled” to “In total 36 organoids from different wells and biological conditions were labeled.”

Reviewer #3 Question 12. Line 454 should state the number of CPUs/threads used. Does the deployment of the method also rely on the utilization of a GPU?

We thank the reviewer for this question. No, the deployment of the method does not rely on the utilization of a GPU. We added the following text to clarify the CPU needs to run parallel jobs.

We changed the following text “As *Cellos* has the ability to process multiple wells of a plate at the same time, the time used to investigate 60 wells was similar to the one used for one well” to “In addition, the analysis of multiple wells was parallelizable, with the time for 60 wells processed by 60 CPUs being similar to the time for one well with one CPU.”

This information is also later clarified in the discussion section “altogether, when pre-processing and morphological feature characterization are included, it took on average ~1.9 hours with CPU efficiency of 91.2% and ~100 GB of computational memory to segment 550 organoids, and ~1.4

hours with CPU efficiency of 82.31% and 6.86 GB of computational memory to segment 4837 nuclei.”

Reviewer #3 Question 13. Line 482, the ability to infer clonal heterogeneity is not something that is intrinsically enabled by Cellos alone and can be achieved using any method that accurately segment cellular regions. Consider revising wording.

The line that the reviewer refers to says “*Cellos* enables interpretation of these parallel measurements by individually quantifying cells and organoids to elucidate chemical biology and pharmacology effects.” The novelty of *Cellos* is that it can do the segmentation in 3D and in high throughput.

To address the reviewer’s point, we have added this sentence to the discussion: “**Clonal heterogeneity in tumors has also long been studied 2D techniques such as immunohistochemistry.**” and we updated the sentence “*Cellos* enables interpretation of these parallel measurements by individually quantifying cells and organoids to elucidate chemical biology and pharmacology effects.” to “***Cellos* facilitates interpretation of these parallel measurements in organoids by individually quantifying and analyzing organoids and their cells in 3D to elucidate growth and pharmacology effects.**”

Reviewer #3 Question 14. Line 508. Avoid the use of superlatives. There have been multiple other papers that have presented a scalable single cell analysis approach for organoids/spheroids (for example <https://www.liebertpub.com/doi/full/10.1089/adt.2015.655>).

To address your comment, we have changed the following text “In conclusion, we report the first high-throughput-compatible pipeline for 3D organoid and nuclei segmentation, as well as morphological quantification.” to “**In conclusion, we report a high-throughput-compatible pipeline for true 3D volumetric organoid and nuclei segmentation, as well as morphological quantification on images with numerous organoids**”.

The paper mentioned above (Sirenko et al.,2015) is an important paper that “proposed methods that can increase performance and throughput of high-content assays for compound screening and evaluation of anticancer drugs with 3D cell models”, including studies of a library of 119 approved anticancer drugs. For clarity, we note some of the distinguishing points of *Cellos*. Sirenko et al described a “maximum projection algorithm that combines cellular information from multiple slices through a 3D object into a single image”, while *Cellos* uses the whole volumetric 3D image and therefore more accurately retains volumes and cell spatial relationships. Moreover, *Cellos* is applicable to multiple organoids per well while Sirenko et al analyzed one organoid/spheroid per well.

Reviewer #3 Question 15. Line 618, what normalization was used (z, an external reference)?

We edited the methods section to clarify the normalization approach.

“In detail, each individual channel of each organoid is normalized before the nuclei segmentation step. This normalization is percentile based and is part of the StarDist method⁴⁵.”

REVIEWERS' COMMENTS

Reviewer #1 (Remarks to the Author):

The manuscript has been improved as suggested.

Reviewer #2 (Remarks to the Author):

I believe the authors have addressed our comments appropriately and have gone to great lengths to enhance the study.

Reviewer #3 (Remarks to the Author):

The authors have satisfyingly addressed all concerns that were raised by myself and I have no remaining major concerns with respect to the publication of this article. However, I did notice some small language errors (such as PDX being defined as Patient Delivered Xenografts at line 94 as opposed to Patient Derived Xenografts). Accordingly, I still recommend copy editing or other professional editing services be obtained before final publication to verify the precision/correctness of the language.